# Embedding of HIV Egress within Cortical F-Actin

**DOI:** 10.3390/pathogens11010056

**Published:** 2022-01-03

**Authors:** Anupriya Aggarwal, Alberto Ospina Stella, Catherine C. Henry, Kedar Narayan, Stuart G. Turville

**Affiliations:** 1The Kirby Institute, University of New South Wales, Sydney, NSW 2052, Australia; aaggarwal@kirby.unsw.edu.au (A.A.); aospinastella@kirby.unsw.edu.au (A.O.S.); 2Center for Molecular Microscopy, Center for Cancer Research, National Cancer Institute, National Institutes of Health, Bethesda, MD 20892, USA; cchenry@mit.edu (C.C.H.); narayank@mail.nih.gov (K.N.); 3Cancer Research Technology Program, Frederick National Laboratory for Cancer Research, Frederick, MD 21702, USA

**Keywords:** HIV-1, HIV-1 Gag, CDC42, IQGAP1, ARP2/3, F-Actin, budding, Diaph2

## Abstract

F-Actin remodeling is important for the spread of HIV via cell–cell contacts; however, the mechanisms by which HIV corrupts the actin cytoskeleton are poorly understood. Through live cell imaging and focused ion beam scanning electron microscopy (FIB-SEM), we observed F-Actin structures that exhibit strong positive curvature to be enriched for HIV buds. Virion proteomics, gene silencing, and viral mutagenesis supported a Cdc42-IQGAP1-Arp2/3 pathway as the primary intersection of HIV budding, membrane curvature and F-Actin regulation. Whilst HIV egress activated the Cdc42-Arp2/3 filopodial pathway, this came at the expense of cell-free viral release. Importantly, release could be rescued by cell–cell contact, provided Cdc42 and IQGAP1 were present. From these observations, we conclude that a proportion out-going HIV has corrupted a central F-Actin node that enables initial coupling of HIV buds to cortical F-Actin to place HIV at the leading cell edge. Whilst this initially prevents particle release, the maturation of cell–cell contacts signals back to this F-Actin node to enable viral release & subsequent infection of the contacting cell.

## 1. Introduction

Actin is a major component of the cellular cytoskeleton and is present in both monomeric globular (G-actin) and polymeric filamentous (F-actin) forms in all eukaryotic cells. Specifically in human leukocytes, actin accounts for over 10% of the total protein content and is a prerequisite for many pathways involved in communication of the immune response, such as chemotaxis of leukocytes through the formation of supramolecular structures like the immunological synapse, a fundamental structure driving the primary immune response. While cells encode a wide range of proteins that mediate F-actin remodeling, the critical ability to seed or ‘nucleate’ F-actin from the monomeric G-actin pool is limited to only a few protein families. The two major classes of cellular actin nucleators are the Arp2/3 complex and formins. The Arp2/3 complex is composed of seven different subunits and allows formation of branched actin networks through nucleation of a branch filament from an existing mother filament at an angle of 70 degrees [1]. In contrast, formins are large multidomain proteins that drive nucleation and/or elongation of unbranched linear actin filaments. The activity of cellular Arp2/3 and formins is tightly regulated by a complex network of signalling pathways that primarily rely on the molecular switch properties of Rho-GTPases, such as Rac1 and Cdc42, for their activation [2].

HIV infection and spread proceeds primarily in CD4 positive leukocytes of our immune system. Viral spread can be observed at two levels. Firstly, free virus release from infected cells, with virions travelling in a cell-free form until encountering a new target cell to infect. Secondly, HIV budding that occurs directly at sites of cell–cell contact. The supramolecular structure that enables the latter and highly efficient cell–cell viral transfer is referred to as the virological synapse (VS) [3,4]. In both cases, viral budding proceeds at the plasma membrane (PM) of infected cells and is initially driven by oligomerisation of the HIV structural protein Gag [5] and culminates with HIV particle abscission mediated by cellular proteins of the endosomal sorting complexes required for transport machinery (ESCRT) [6]. Several F-actin structures have been previously observed in association with HIV assembly and higher-level Gag oligormerisation. These include the temporal formation of F-Actin asters/stars that appear just underneath the PM prior to particle release [7] and assembling HIV particles decorating the tips of finger-like filopodial structures [8,9,10]. It is, however, unclear how these events are mechanistically connected and how coupling to the F-actin cytoskeleton benefits HIV release/spread. This can be considered at two-levels: firstly, how does F-Actin regulation influence the assembly and release of cell free virus in infected cells? For instance, do cortical F-Actin structures facilitate HIV assembly and release at the PM as observed for other viruses [11]? Secondly, how is F-actin regulated at cell–cell contacts involving HIV infected cells? Several studies have shown that functional actin dynamics are required for cell–cell viral transfer [4], yet, how HIV assembly and release are spatiotemporally coordinated during this process has not been clarified mechanistically. 

With our primary aim to determine the role for F-Actin in cell-free HIV egress and cell–cell viral transfer, herein we peeled back the complexity of F-Actin regulation in leukocytes by successive depletion and/or knockout of key actin nucleators and other associated proteins that regulate their activity. In doing so, we biased the formation of different cortical F-Actin structures such as filopodia and lamellipodia. Herein we define these structures as outlined by Mattila and Lappalain, i.e., Filopodia are cylindrical finger like protrusions approximately 100–300 nm in diameter and up to 1 µm to 10 µm or more in length, whereas lamellipodia are thin (100 nm to 200 nm thick) sheet/viel like cortical F-Actin protrusions [12]. Whilst the regulation of lamellipodia is well understood and primarily depends on branched F-actin nucleation by the Arp2/3 complex downstream of Rac1 and its effector Wave2 [13,14], various models have been proposed for filopodia formation; increasing evidence suggests this process may be cell-type specific [15]. Importantly, little is known about the mechanism of filopodia formation in cells of hematopoietic lineage, despite the fact that filopodia play important roles in immune cell function. 

Using a combination of live cell imaging, focused ion beam scanning electron microscopy (FIB-SEM), virion mass spectrometry and viral infection assays, we observed the influence of HIV on cortical F-Actin at several levels. First, FIB-SEM revealed HIV budding to be relatively enriched in areas of high positive membrane curvature within Arp2/3-dependent cortical F-Actin structures, including filopodia and lamellipodia. Second, virion mass spectrometry identified a cortical F-Actin signaling node comprising of the Arp2/3 related GTPases Rac1 and Cdc42 and their binding partner, the scaffolding protein IQGAP1. Finally, while depletion of several dominant F-actin regulators was observed to affect free virus release, cell–cell viral transfer was only significantly impaired in cells depleted of Cdc42 or IQGAP1. Collectively, these observations support a dominant role of the GTPase Cdc42 and IQGAP1 in the final stages of viral egress and cell- cell spread. In this setting, we propose HIV manipulation of the Cdc42/IQGAP1 node to be important at two levels: firstly, it enables HIV to be embedded and retained in Arp2/3- dependent leading edge structures that are important during pre-synaptic events. Secondly, as the vs. is engaged and matures, the same regulators likely coordinate F-Actin dynamics to enable conditions that facilitate final viral particle release. 

## 2. Results 

### 2.1. Moulding Cortical F-Actin through Formin and Arp2/3 Depletion 

A physical association of HIV with F-actin structures has been previously observed in all major HIV primary target cell types [9,16]. In infected CD4+ T-cells and dendritic cells, this manifests in the form of HIV-Filopodia, which are F-actin rich finger-like structures with HIV assembly observed at their tips [9]. Since these structures are more prominent on dendritic cells and similarly enriched in U937 cells [9,17,18], this latter promonocytic cell line provides an ideal model to dissect the link between F-Actin and HIV assembly in the specific context of myeloid hematopoietic cell lineages. Given the proposed role of Arp2/3 and formins in filopodia formation in other cell types, we initially focused on these key actin regulators for depletion. However, while Arp2/3 is ubiquitously expressed in eukaryotic cells, there are at least 15 different formins in vertebrates [19]. Since Diaph1, Diaph2 and FMNL1 are the ones most abundantly expressed in leukocytes, we tailored our initial shRNA screening to depletion of these formins. Disruption of filopodial networks was assessed by measuring filopodial abundance (average number of filopodia per cell) and length (average filopodial length measured from the PM to the tip). To this end, we used several imaging techniques at increasing levels of resolution, including; (i) live cell imaging, (ii) fixed cell fluorescence imaging followed by 3D deconvolution, and (iii) the power of correlative FIB-SEM to finely resolve F-Actin structures closer to the PM. In brief, FIB-SEM represents a method where iterative cycles of finely tuned ion abrasion milling are followed by high-resolution scanning electron microscopy of heavy-metal stained, resin-embedded cell samples [20,21]. The end result is the recording of a stack of 2D back-scatter electron images, which are then processed and converted to a 3D image volume, typically at ~10 nm pixel sampling (Figure 1B–E). This method provides a powerful imaging tool for cell biology and virology, as it gives users the ability to resolve nanoscale ultrastructural features in cellular samples that may appear in association with viral particles [22].

Initial experiments revealed filopodial lengths to be dependent solely on the formin Diaph2 and not the other leukocyte-enriched formins (Appendix A). Silencing of Diaph2 by shRNA achieved >95% depletion at the protein level (Appendix A); in these cells, cortical F-Actin coalesced into a network of abundant and short (1 to 3 µm) filopodia (Figure 1D,G; Appendix A). Since we could confirm this phenotype in CRISPR/Cas9-generated and clonally expanded Diaph2 homozygous knockout cells, our observations suggest that filopodial length, but not seeding, is dependent on Diaph2. Subsequent shRNA codepletion of other expressed formins in addition to Diaph2 also did not disrupt this short filopodial network (Appendix A). Therefore, to test if the shorter but more abundant filopodia were Arp2/3-dependent, we disrupted both Diaph2 and the Arp2/3 complex by shRNA. Co-depletion led to short filopodia converging into an extensive lamellipodial network (Figure 1A,E,H; Appendix A). To conclude, we could readily control cortical F-Actin within this leukocyte landscape and generate three unique cell types with a continuum of cortical F-actin structures: (i) long filopodia, (ii) abundant short filopodia, and (iii) an extensive network of large lamellipodia. Furthermore, our observations indicate that seeding of filopodia in myeloid cells requires Arp2/3-mediated actin nucleation, whereas filopodial elongation is dependent on the formin Diaph2 (Figure 1F–H).

### 2.2. The Influence of Shifting F-Actin Structures on the Location of HIV Budding

In the context of HIV infected cells, we used our high-resolution imaging approaches to probe for a possible link between HIV assembly and specific F-Actin structures and/or pathways in leukocytes. Previously, we had observed live cells with long filopodia to have significant numbers of HIV positive tips [9] and readily concluded that, in untreated cells, HIV assembly is enriched to this site. However, detection of HIV buds in cells with short filopodia (i.e., depleted of Diaph2) was constrained by the inability to resolve F-Actin structures proximal to the PM by fluoresecence microscopy alone. Thus, we applied FIB–SEM imaging to HIV-infected Diaph2^−ve^ cells (short filopodia) and observed HIV buds in routine association with the tip and sides of these structures (Figure 2A,B,E,H). In cells with prominent lamellipodia (Diaph2^−ve^Arp2/3^−ve^), FIB-SEM imaging revealed abundance of HIV buds along the ridges of lamellipodia (Figure 2F,K; Appendix A). Therefore, one common feature of each F-Actin structure was the appearance of HIV preferentially in areas of positive membrane curvature. Given the large topological differences between filopodia and lamellipodia, we assessed viral-bud density by accounting for the surface area available for budding within each distinct F-Actin structure. This revealed HIV buds to be significantly enriched in areas of high positive membrane curvature (Appendix A; Figure 2G–L); lamellipodial ridges and filopodia were the most active areas of viral assembly, with a distinct preference for the latter (filopodia tips outscored lamellipodial ridges by 5-fold when surface area was considered). This observation supports two potential mechanisms. First, HIV assembly is facilitated by areas of positive curvature or alternatively, HIV assembly recruits/influences cellular protein(s) at the PM that can lead to positive curvature.

### 2.3. Filopodia Dominated by the Formin Diaph2 Present Positive Curvature at the Plasma Membrane but Exclude Assembling HIV Particles

To assess whether strong positive membrane curvature alone was sufficient to position HIV buds at filopodial tips, we induced long filopodia using a constitutively active (C/A) mutant of Diaph2. Diaphanous-related formins exist in an autoinhibited conformation mediated by the interaction between their N-proximal inhibitory domain (DID) and C-terminal autoregulatory domain (DAD) [23]. Disruption of this autoinhibitory state can be achieved by formins binding to Rho-GTPases or, as in our case, by deletion of their C- terminal DAD domain [24]. Importantly, in both cases, the central actin polymerization domain of the formin is rendered constitutively active.

If HIV assembly was directly promoted by positive membrane curvature, filopodia induced by Diaph2^C/A^ would incorporate assembling viral particles. However, while Diaph2^C/A^ expression readily induced the formation of long straight filopodia with Diaph2 accumulating at the filopodial tips (Figure 3A & Appendix A), in HIV infected cells, we also observed complete exclusion of HIV particles from the tips of these structures (Figure 3B,C & Appendix A). Therefore, the strong membrane curvature in filopodial tips alone is not sufficient to recruit HIV assembly to this region. Since long filopodia in WT cells are routinely HIV positive, whereas straight C/A Diaph2 driven filopodia are not, it is unlikely that formins represent the link of HIV to the F-actin cytoskeleton. We then turned our attention to the Arp2/3 complex, given that previous observations propose this as the dominant F-actin nucleator at the cell cortex, with formin activity being restricted to filament elongation post F-actin nucleation [25]. To confirm if Arp2/3 was the dominant filopodial nucleator in WT vs. Diaph2^C/A^ cells, we immunostained filopodia for Arp2/3, and examined the footprint of this nucleator along the filopodial body and tip. In both cell types, the filopodial bases (3 μm from the membrane) were all Arp2/3 positive (Figure 3D,E). In contrast, the filopodial tips of Diaph2^C/A^ cells were negative for Arp2/3 antigen (Figure 3D), whereas Arp2/3 was frequently observed along the entire shaft and at the tip of wildtype filopodia (Figure 3E). To quantify the extent of Arp2/3 tip exclusion, we measured the distance from the tip of filopodia to the first detectable Arp2/3 signal and observed a significantly greater distance of Arp2/3 from the filopodial tip in Diaph2^C/A^ cells relative to WT cells (6.2 μm vs. 1.4 μm; *p* > 0.0001, *n* = 50) (Appendix A). In summary, by mapping the HIV budding sites at high resolution, we could reach several conclusions. Firstly, HIV buds primarily enrich to cortical F-Actin structures with positive curvature. Secondly, positive curvature and/or Diaph2 activity alone are not responsible for the enrichment of HIV buds to these sites. Finally, Arp2/3-dependent cortical F-actin structures are primarily HIV positive.

### 2.4. HIV Gag Can Influence Arp2/3 Dependent F-Actin Pathways

Since HIV assembly at the PM is primarily driven by HIV-Gag, we turned to strategic Gag mutagenesis in an attempt to resolve the link of HIV assembly with cortical F-Actin structures. The HIV Gag mutant panel covered several well characterised mutants that could maintain HIV particle assembly, as well as binding to membrane Phosphatidylinositol (4,5)-bisphosphate (PIP2). Deletion of HIV Gag p6 and mutagenesis of the PTAP motif in p6, was used to block the recruitment of TSG101 and related ESCRT proteins involved in viral particle abscission (Figure 4A,B). We also deleted the Nucleocapsid (NC) domain, as this has been previously proposed to mediate the interaction between HIV-Gag and F-actin [26]. However, since NC is required to facilitate higher order oligomerisation of Gag [27], we replaced NC with the leucine zipper (LZ) domain from the *Saccharomyces cerevisiae* GCN4 protein (Figure 4C), as this rescues Gag oligomerisation and ensures particle assembly proceeds in the absence of NC [28]. Finally, given the enrichment of HIV buds on F-Actin structures with positive curvature, we further generated two HIV Gag capsid mutants P_99_A and EE_75_,_76_AA (Figure 4D,E), both of which inhibit Gag curvature at the PM but not high order Gag ligomerization [29,30]. As Diaph2 cannot recruit HIV to F-Actin and depletion of Diaph2 actually enriched HIV-positive filopodia (Appendix A), we utilised Diaph2^−ve^ cells and simply scored the number of filopodia per cell that were HIV positive for each viral Gag mutant. Using this approach, we observed no significant difference in viral filopodia when deleting p6, the PTAP motif in p6 or NC (Figure 4F). However, when using the P_99_A and EE_75_,_76_AA HIV capsid mutants (HIV curvature mutants), we observed Diaph2-depleted cells to not only lack any evident HIV buds at the PM but also their characteristic short filopodia. Instead, these cells resembled the lamellipodial phenotype observed in Diaph2^−ve^Arp2/3^−ve^ co-depleted cells (Figure 4G–I). This is consistent with HIV Gag curvature mutants acting as dominant negatives for the Arp2/3-dependent short filopodial pathway. To further test that HIV curvature mutants were specifically interrupting Arp2/3 F-actin pathways and not broadly influencing all pathways that may lead to filopodial formation (e.g., Formin-induced filopodia), we infected Diaph2^C/A^ cells with these mutants. In this setting we observed an ability of Diaph2^C/A^ to rescue filopodia formation relative to Diaph2 depleted cells infected with the above mutants (Figure 4F; Appendix A). Thus, filopodial pools nucleated by Arp2/3 were most affected by HIV curvature mutants, which further supports the hypothesis that HIV assembly primarily influences elements of Arp2/3 F-Actin nucleation pathway.

### 2.5. The HIV Proteome Reveals a GTPase Node Associated with Arp2/3 F-Actin Regulation

HIV has been previously observed to incorporate F-Actin, various actin nucleators and numerous upstream/downstream regulators within virions [31,32,33]. Thus, we turned to mass spectrometry analysis of purified virions to observe the footprint of cytoskeletal proteins that are present at HIV assembly sites. For this analysis we also leveraged the three distinct F-Actin cell types generated above (i.e., long-filopodia, extensive short filopodia and large lamellipodia) as across each cell type they shared the feature of HIV buds being enriched in positively curved F-Actin structures. Using this approach, we identified the three canonical Rho-GTPases RhoA, Cdc42 and Rac1. The latter two are well known as major regulators of the Arp2/3 complex, of which we also identified various subunits in the virion proteome. These regulators were observed across all viral proteomes, irrespective of producer cell type (Figure 5A,B), thus suggesting an important role of the Arp2/3 complex in viral egress. IQGAP1, a large scaffolding protein which plays major but poorly understood roles in cellular actin regulation, was also identified. HIV virions also acquired members of the integrin and cadherin families (Figure 5A,B; see nodes 3 and 4, respectively). These proteins, which are involved in cell–cell adhesion, are connected to the cortical F-actin cytoskeleton both physically and via signaling pathways [34]. Of interest was a depletion of the cadherin node in (Figure 5A; node 4), as well as an increase in Arp2/3 and Cdc42 content (Figure 5A; node 1) in virions produced by Diaph2-deficient cells. The latter observation is not only consistent with HIV assembly preferentially proceeding alongside short Arp2/3-dependent filopodia (as observed by FIB-SEM), but also suggests that these structures are dependent on Cdc42, which is a well-known filopodial regulator [2]. Of note, the observed decrease of Arp2/3 components in virions produced in Diaph2 and Arp2/3 co-depleted cells (Figure 5B, node 1) is both expected and consistent with depletion of these proteins at the cellular level (Appendix A).

### 2.6. HIV Exploits the Cdc42-Arp2/3 Filopodial Pathway to Position Virus at Cell–Cell Contacts

Since Cdc42 is an important regulator of Arp2/3 and a master regulator of filopodia; it was incorporated at higher levels in virions from our Diaph2^−ve^ cells (more abundant short filopodia), we next targeted this protein for depletion. As a functional control, we targeted the homologous Rho-GTPase Rac1, best known for its role in lamellipodial regulation. We also investigated the scaffolding protein IQGAP1, in which; (i) is a binding partner and effector of both Cdc42 and Rac1 [35], and (ii) plays an increasingly recognized role in actin cytoskeleton regulation [36], and (iii) was consistently incorporated in virions in our experiments (Figure 5). While we succeeded in generating a viable Cdc42 homozygous knockout cell line using CRISPR/Cas9 (Appendix A). Attempts at knocking out Rac1 led to multinucleated cell populations with reduced viability, which is consistent with previous reports of Rac1 being an essential gene [37]. To circumvent this, we partially depleted Rac1 by shRNA and also generated a Wave2^k/o^ cell line (Appendix A), since Wave2 is the main downstream effector of Rac1 in F-actin regulation [38]. While obtaining homozygous IQGAP1 knockout clones via CRISPR-Cas9 proved challenging, we were able to establish a line stringently depleted of IQGAP1 using shRNA (>99% depletion at the protein level, Appendix A).

Initial Rac1 depletion via shRNA revealed a greater frequency of filopodia in infected cells; secondly, the generation of significantly longer and thicker filopodia when cells were infected (Figure 6B vs. Figure 6A). Similarly, WAVE2^k/o^ cells infected with HIV had greater propensity to form filopodia (two-fold), and these were significantly longer and thicker compared to WT cells (Figure 6C vs. Figure 6A and Appendix A), but also uninfected WAVE2^k/o^ cells (Appendix A). Together, these observations suggest that HIV infection stimulates a pathway of filopodial formation that is unchecked in Rac1^−ve^ and WAVE2^k/o^ cells, where the lamellipodial F-actin arm is disabled. Given the known role of Cdc42 in filopodia formation and its competing nature with the Rac1 pathway, we turned our attention to this Rho-GTPase. Importantly, Cdc42^k/o^ cells were devoid of filopodia and coalesced cortical F-Actin into prominent lamellipodia, with no evident influence on F-actin when cells were HIV infected (Figure 6D). Since IQGAP1 has been previously reported to articulate Cdc42 signaling to the cytoskeleton [36], we also assessed the role of this regulator in the filopodial context. IQGAP1-deficient cells displayed a collapse in filopodial lengths (Figure 6E), with maintenance of HIV at the tip of remaining filopodia, similar to that observed in Diaph2-depleted cells. We therefore concluded that IQGAP1 can influence filopodial networks but, like Diaph2, is not required for the seeding of filopodia. To summarize our combined observations from mass spectrometry, gene silencing and high-resolution imaging, reveal that HIV infection augments a pathway of filopodia formation, which is most evident when the lamellipodial regulators are inactivated. In contrast, removing Cdc42 completely blocked filopodia formation in a manner similar to Arp2/3 and Diaph2 co-depletion, whereas depletion of IQGAP1 or Diaph2 led to shorter filopodia. Together, our data indicated that HIV-assembly hijacks a cellular pathway that is dependent on Cdc42-Arp2/3 F-actin nucleation for filopodial seeding and IQGAP1/Diaph2 for filopodial elongation, in order to position itself at the tips of long filopodia.

### 2.7. HIV Cell–Cell Transfer Is Dependent on an Intact Cdc42-IQGAP1-Arp2/3 Pathway

Given the continuum of phenotypes observed in our abovementioned observations, we tested their impact on the late stages of the viral life cycle in the context of viral spread. For free virus release, we enumerated HIV particles accumulating in the supernatant as a measure of budding. As HIV spread can also proceed through direct cell–cell contacts, we further tested the ability of HIV to spread cell to cell by coincubating infected donor cells with permissive target cells. Using these approaches, we could determine if the generic lack of a cortical F-Actin structure or a specific F-Actin pathway is essential for HIV budding and/or cell–cell transfer.

Co-depletion of Diaph2 and Arp2/3, but not Diaph2 alone, led to a relatively small but statistically significant reduction in cell–cell HIV transfer (Figure 7B). To confirm the role of Arp2/3 in this process we attempted to disable the complex with the small-molecule inhibitor CK-666. However, use of this compound led to extensive toxicity at doses of >200 µm while no effect on cell–cell HIV transfer was observed at lower doses tested (data not shown). Instead, to explore which Arp2/3-dependent pathways were involved in this process, we independently disrupted the Rac1-WAVE2 pathway (lamellipodia) and the Cdc42-IQGAP1 pathway (filopodia). Both impaired free HIV budding, as indicated by significantly lower viral particle counts in the supernatant from cells depleted of these regulators, compared to untreated cells (Figure 7B). Furthermore, lack of budding was not associated with decreased viability of each cellular clone or lack of HIV Gag expression (Appendix A respectively); impaired release of free HIV did not predict outcomes for cell–cell HIV transfer. For cells with disabled Rac1/WAVE2 (Rac1^−ve^, Wave2^k/o^ cells) cell–cell HIV transfer persisted (Figure 7A,B), despite the decreased free virus budding ability. In contrast, disruption of the Cdc42/IQGAP1 axis (Cdc42^k/o^ and IQGAP1^−ve^ cells) impacted both HIV budding and cell–cell transfer (Figure 7A,B). Of note, we also initially explored a potential role of RhoA, as it was identified alongside Cdc42 and Rac1 in the virion proteome (Figure 5A,B). However, given a complete lack of morphological or cell–cell transfer phenotype in RhoA-depleted donor cells (data not shown), this was not pursued further.

The above observations suggest that while normal actin dynamics are important for free virus release, Cdc42 and IQGAP1 are specifically required for cell–cell HIV transfer, whereas Rac1/Wave2 are not. To further test this hypothesis in the setting of primary CD4 T cell targets, we focussed cell–cell transfer assays with disruption of the Rac1- WAVE2 pathway vs. disruption of Cdc42-IQGAP1. In this setting, we further tested the efficiency of cell–cell spread by limiting dilution of the infected donors into primary CD4 T cell co-cultures. Using this approach, we observed almost complete loss of cell–cell HIV transfer in Cdc42^k/o^ and IQGAP1^−ve^ clones, whereas cell–cell transfer persisted in Rac1^−ve^, Wave2^k/o^ clones, albeit slightly lower than in WT cells (Figure 7C). In cells lacking filopodia (CDC42 and IQGAP1), one immediate mechanism for lack of viral transfer could be the culmination of a limited contact capacity with the cells immediate microenvironment. To test this hypothesis, we enumerated accumulative cell to cell contacts (Figure 7D) and later target cell engagement (Figure 7E) in wild type, IQGAP1^−ve^ and WAVE2^k/o^ cells. IQGAP1^−ve^ cells were observed to have significantly lower overall contacts and also engaged fewer cell targets. Whilst this lowered ability of cells without filopodia to participate in cell–cell transfer, it does not address the paradox of persistent cell–cell HIV transfer, despite reduced viral budding in cells with augmented filopodia. To resolve this further, we observed later interactions WAVE2^k/o^ cells where filopodial networks are augmented following HIV infection. Filopodia initially persisted in early cell–cell conjugates (Appendix A), yet we routinely observed collapse of filopodial networks immediately preceding vs. formation and HIV-GFP transfer to the opposing target cell (Figure 7F and Appendix A). As a surrogate of filopodial activity, we quantified this as membrane complexity through calculation of cellular circularity. In this setting, cells with extensive filopodial networks were observed to have low circularity, whilst cells with no filopodial activity were observed to have high score in circularity. Measurements of Gag-polarisation over time then established measurements of the seeding of the vs. and release of GFP into the neighbouring target was used to mark the final stage of vs. maturation that culminated in viral transfer. Using this quantification in the representative live cell movie acquisitions (Figure 7F,G), we observe cells engaged in cell–cell contact to approach a circularity of 1 (i.e., Cells collapsing their filopodial networks) just prior to the final stages of viral transfer, as marked initially by Gag polarisation and then subsequently observed in cytoplasmic transfer of GFP to the neighbouring cell (Figure 7G).

## 3. Discussion

The corruption of cortical F-Actin by HIV has remained elusive, with evidence both for and against its role in budding and cell–cell viral spread. Through systematic depletion of various F-Actin regulators, combined with viral mutagenesis and high-resolution imaging, we were able to illuminate the intersection of HIV egress with cortical F-Actin and conclude this is primarily associated with the Cdc42-IQGAP1-Arp2/3 pathway.

Our primary aim herein was to understand how HIV egress was spatiotemporally connected to a continuum of cortical F-actin structures that dynamically regulated in leukocytes. Whilst many prior studies have mapped F-Actin pathways in cell-free systems, the challenge herein was to map F-Actin pathways and how they influenced not only the live virus, but also in a cellular & cytoskeletal setting that was consistent with that of the immune system. Whilst many F-Actin regulators are common across cells, F-Actin regulation in leukocytes is unique and enables a rapidly changing canvas of F-Actin polymers to coordinate their roles in the immune response. Chemotaxis, promiscuous cell–cell scanning and later stable cell–cell contacts are all processes dependent on dynamic cortical F-Actin. How F-Actin influences HIV egress needs discussion into two distinct events. Firstly, HIV’s time at the leading edge of protruding F-Actin structures (target selection) and, secondly, when infected cells engage in longer stable contacts (target engagement).

At the leading edge of F-actin, we observed HIV buds to be enriched where cortical F-actin structures induce strong positive curvature. However, curvature and/or formin activity alone were not sufficient to position HIV at the tips of filopodia. Thus, other actin regulators associated with membrane curvature must be involved. Our observations herein that HIV infection specifically enhances Cdc42-Arp2/3-dependent filopodia, supports a mechanism of action that corrupts this pathway of actin nucleation. While curvature provided by HIV during budding could itself drive filopodia formation (e.g., by direct recruitment/activation of Cdc42/Arp2/3 [39,40], our observations support hijacking of a pre-existing pathway dependent on curvature. We base this reasoning three-fold. Firstly, on uninfected myeloid cells there are similar long filopodia (albeit uncapped with HIV). Secondly most filopodia in infected cells are HIV-capped [9]. If HIV would provide an independent mechanism of filopodia formation, both structures would be expected to coexist. Thirdly and finally, HIV curvature mutants had a dominant negative effect on all filopodia, indicating (i) a functional overlap of the viral and cellular pathways of filopodia formation, and (ii) a critical role of Gag in hijacking of these structures. Recently, Sabo and colleagues observed HIV Gag to directly interact with IQGAP1 [41] This observation is consistent with our observations herein at several levels. Firstly, IQGAP1 binds to Cdc42 and stabilizes it in its active conformation to drive filopodia formation [36]. Secondly, IQGAP1 also facilitates assembly of multiprotein complexes that spatially link Cdc42, Arp2/3 and formins [42]. This is consistent with how HIV is associated with a filopodial structure that is firstly nucleated by CDC42, but secondly elongated by the formin Diaph2. Finally, given the role of positive curvature in filopodia biogenesis, consumption of IQGAP1 by HIV Gag into an area of neutral curvature, is consistent with HIV Gag curvature mutants acting as dominant negative constructs for CDC42-Arp2/3 filopodia.

Whilst others have also reported Cdc42 and Arp2/3 to be critical for the formation of F-actin structures that contribute to cell–cell transfer of HIV from dendritic cells [43] it must be noted there are two distinct differences with respect to these structures. Firstly, they are not filopodia but rather sheet-like F-Actin structures [22,43]. Secondly, these F-Actin sheets form in uninfected cells they can only mediate cell–cell transfer “in trans”. In the study described herein, HIV-Filopodia form in infected cells and participate in cell–cell spread when a donor cell is productively infected (in cis). Overall, the fact that HIV exploits common actin regulators for facilitating HIV transfer both in cis and in trans highlights how critical this biological host-pathogen intersection is for the virus and encourages intervention strategies targeting these common regulators, as this would likely counteract both modes of HIV spread.

Whilst the resolution of pathways that gives birth to HIV filopodia are now becoming clearer, the role of this hybrid viral and cellular structure now needs discussion. Typically, viral-cellular membrane events are associated with viral release. In contrast, we observed HIV’s association with F-Actin to be inhibitory to HIV budding. The most evident is when HIV filopodial networks are formed during Rac1 and Wave2 depletion. The most evident is when HIV filopodial networks are formed during Rac1 and Wave2 depletion. In that setting, the removal of Rac1 would not only bias signalling to CDC42, but with IQGAP1 recruited by HIV Gag, CDC42 would be maintained in an active GTP bound state [36] and, as such, promote IQGAP1 oligermerisation [44]. This is entirely consistent with its role in filopodial formation but also its augmentation by HIV Gag-IQGAP1 during infection. Whilst arrest of viral HIV buds in a F-Actin structure may be a function of distance that this structure projects immature HIV from the membrane, the role of the virus in this setting seems counter intuitive at two levels. Firstly, a virus that cannot undergo absciscion at the membrane cannot subsequently mature [45] and thus cannot contribute to viral spread. Secondly, it is well known HIV particles bud from infected cells. To reconcile both observations, we hypothesise that while most HIV buds at the membrane engage in the budding process, a proportion of these remain in prolonged association with F-actin protrusions, which can indeed indirectly contribute to cell–cell spread. These retained virions may indeed be hard-wired to coordinate the initial pre-synaptic contacts. Yet when cell–cell contacts mature, HIV’s relationship with cortical F-Actin likely undergoes reconfiguration, to allow the inhibitory mechanisms observed during pre-synaptic events to be overcome and give way to cell–cell viral transfer.

Shadowing and highjacking the Cdc42-IQGAP1-Arp2/3 actin regulatory axis is not a unique feature of HIV. Nascent viral buds have also been observed at the tips of filopodia-like structures for other types of viruses [46,47,48,49,50,51]; numerous intracellular pathogens are known to exploit Rho-GTPases and the unique ability of the Arp2/3 complex to promote formation of specialized cortical F-actin membrane protrusions that facilitate cell–cell infection spread [52,53,54,55,56]. Similarly, IQGAP1 is a prominent target of microbial manipulation and this is closely related to its ability to modulate the actin cytoskeleton (reviewed in [57]). Several viruses bind IQGAP1 either directly (via interactions with the viral matrix protein) or indirectly (via common binding partners) and this has important consequences for viral assembly, budding and/or pathogenesis [11,48,58,59,60,61] For HIV, recent studies by Sabo and colleagues [41], have observed biochemically IQGAP1 interactions with NC & p6 elements of HIV Gag. In this study they support a role for IQGAP1 in negative regulation of HIV Gag trafficking and subsequent HIV-budding [41]. Whilst our observations readily support a role for IQGAP1 in influencing HIV budding, we did not observe this to be a consequence of negative regulation of Gag trafficking and docking to the membrane. For instance near complete removal of IQGAP1 did not increase viral budding and egress, but rather led to inhibition thereof. In light of our observations herein and those recently published [41], it can be concluded that given IQGAP1 is a scaffolding protein with many binding partners, the fate of IQGAP1 bound HIV Gag may have many different fates. Furthermore, these fates will depend on each cell type it is expressed in and what functions that cell type maybe engaged over the time the cells were sampled. Importantly, we do readily support a role for IQGAP1 in the viral life cycle and this readily supports recent observations by this team.

In terms of HIV spread, free viral particle release was susceptible to inhibition of both lamellipodial and filopodial regulators. In contrast, cell–cell spread of HIV was mainly dependent on Cdc42 and IQGAP1 but showed tolerance to depletion of Rac1/Wave2, despite a similar impact of all regulators on free virus budding. Together, these observations suggest that long filopodia specifically contribute to cell–cell HIV spread, whereas lamellipodia are less important for this process. This is consistent with findings from other enveloped viruses where filopodia have been associated with the ability to mediate cell–cell viral transfer [62,63,64,65,66]. Indeed, we have previously observed HIV-filopodia to mediate hundreds of contacts per hour between relevant primary HIV target cells, with filopodial activity often preceding vs. formation [9]. The latter is in agreement with previous observations that filopodia and/or dendrites may commonly serve as precursors for biological synapses [67,68,69,70,71]. It has also been proposed that, when synapses mature, filopodia must be “suppressed to allow a smooth and broad cell–cell interface” [71]. This is consistent with our observations herein that early HIV donor-target cell contacts are characterized by abundant filopodial activity, whereas filopodia are often lost during late stages of vs. formation, when polarization and transfer of Gag is most evident. These remarkable cell-shape changes reveal that vs. progression involves extensive cytoskeletal remodeling and suggests a clear switch of actin-manipulation strategy as the synapse matures. Loss of filopodia likely requires inactivation of Cdc42, which could be important to allow synchronised budding and large-scale viral release at the VS, in a similar manner to how Cdc42 inactivation promotes mechanistically analogous (i.e., ESCRT-dependent) abscission events during late-stage cytokinesis. Thus, IQGAP1 may serve as a scaffolding center that enables coordination and cross-signaling of F-actin remodelling and abscission events during both cytokinesis and the VS. However, as with other synapses, maturation of the vs. is a temporally and mechanistically complex process, and further studies will be required to fully elucidate the mechanisms involved.

## 4. Materials and Methods

### 4.1. HIV Plasmid Constructs

Plasmid constructs used herein are all based on CCR5 using pNL43AD8^ENV^, unless otherwise indicated (Appendix A). All details regarding GFP carrying HIViGFP and HIViGFP^ENV−ve^, including the insertion of GFP in Gag polyprotein, have been previously described [9]. HIV nucleocapsid mutant (gag-p7^−ve^-LZ) was generated using a two-step cloning strategy wherein a 861 bp fragment carrying the HIV Capsid domain, P2 spacer region and the leucine zipper domain of yeast transcription factor GCN4 (herein LZ) was amplified from plasmid pRR546, a kind gift from Dr. A Rein (NCI, Bethesda, MA, USA), [28] and shuttled into HIViGFP using *XbaI*/*ApaI* cut sites. The intermediate plasmid (HIV^LZ1stGen^) thus generated had most of the NC domain replaced with the LZ domain, except for a 75 bp region proximal to the p6 domain. To remove this fragment and have the LZ domain contiguous with the p6 domain, primers were designed to amplify a 760 bp fragment from HIViGFP containing the p6 domain and Pol region proximal to Sbf1 restriction site, while simultaneously introducing an *Apa1* restriction site at the 5′end. The resulting amplicon was then shuttled into HIV^LZ1stGen^ using *ApaI*/*SbfI* cut sites to generate gag-p7^−ve^-LZ. Capsid mutant constructs gag-Capsid-EE75,76AA and gag-Capsid-P99A were generated by site directed mutagenesis, wherein the reverse primers were designed to include the desired mutations. For gag-Capsid-EE75,76AA, a 320 bp fragment downstream of GFP was amplified from HIViGFP with the mutagenesis primers carrying the respective mutations (EE75,76AA) and re-cloned into the parental construct using *XbaI*/*SphI* unique sites, while, for gag-Capsid-P99A, a 370 bp fragment downstream of GFP was generated with the mutagenesis primers carrying the point mutation, P99A, and shuttled back using *XbaI*/*SpeI* sites. HIV p6 deletion mutant gag-p6^−ve^ was synthesised by cloning a 1.5 kb fragment from plasmid L1-term, courtesy of Dr. Eric Freed, NIH, USA, into HIViGFP using *SpeI*/*SbfI* restriction sites. The construct L1-term carries a stop codon at the start of p6 domain of Gag in pNL43 and has been described in detail elsewhere [72]. pHIVNL43IRESeGFP (Courtesy of Dr. Paul Cameron, Doherty Institute, Melbourne) expressed eGFP in the Nef open reading frame, followed by an IRES element and the intact HIV Nef open reading frame. All of the above clones were sequenced, verified and transfected into HEK F293T cells (Invitrogen) as previously described [9].

### 4.2. Virus Production and Infections

In order to examine Gag mutations in context of infectious virus, we generated viruses capable of single round of infection by ‘rescuing’ HIViGFP Gag mutants with a 2nd generation lentiviral packaging construct psPAX2 (courtesy of Didier Trono through NIH AIDS repository). The latter construct expresses wild type HIV Gag and Gag-Pol under a CMV based promoter and enables HIV genomes encoding of the indicated Gag mutants to be packaged into viral particles (consisting of both mutant and WT Gag proteins) and importantly entering and infecting cells in a single round of infection. As psPAX2 only supplies WT Gag and Gag-Pol at the protein level, following infection, only the products of the mutant HIV genome are expressed (i.e., only the Gag mutant protein). To generate ‘rescued’ HIViGFP Gag mutants, viral stocks for U937 infections, the mutant constructs, were co-transfected with psPAX2 and also with the VSVg plasmid pMD2.G (Addgene; courtesy of Dr. Didier Trono), at a molar ratio of 2:1:1 to increase the infection rate. Transfections were done in HEK F293T cells using polyethylenimine (at 1 mg/mL, pH 7.0), as described previously [9]. For cell-to-cell transfer assays, pHIVNL43IRESeGFP was used and was produced by polyethylenimine transfections of HEK F293T cells. TZM-bl indicator cell line (courtesy of the AIDS reagent repository) was used for virus titering and analysis, as previously described [9].

### 4.3. Lentiviral Constructs and Production:

#### 4.3.1. Diaph2^C/A^ Mutant Construct

Lentiviral plasmids expressing the constitutively active form of Diaph2 (pLVXdeltaDADmcherry) fused to mCherry was synthesised using the *XhoI*/*ApaI* restriction sites. The 30 bp autoregulatory domain (DAD) of Diaph2 with the consensus sequence DET(G/A)(V/A)MDXLLEXL(KIR/Q)X(G/A)(S/G/A)(A/P) spans aa 1051–1081 at the C-terminus and was removed from Diaph2 cDNA during cloning. Forward and reverse primers carrying the above restriction sites were designed to amplify a 3 kb fragment from Diaph2 MGC human cDNA clone (Dharmacon, Lafayette, CO, USA) and the amplicon encompassing the entire cDNA sequence minus the DAD domain was cloned into pLVX-mCherry-N1 (Clontech, Mountain View, CA, USA) to generate a lentiviral expression vector pLVXdeltaDADmcherry. The plasmids were sequence verified and lentiviral particles were then generated using the helper plasmid, psPAX2 and pMD2.G, as previously described [9].

#### 4.3.2. Lentiviral shRNA Vectors

shRNA sequences for each gene target were obtained from The RNAi Consortium (TRC) library database and sequences with high adjusted scores (2 shRNA sequences per gene) were selected for synthesis (Appendix A). Oligonucleotides (both sense and antisense) carrying *Age1*/*EcoR1* sticky ends, hairpin loop sequence and shRNA sequence were synthesised (IDT technologies, Coraville, IA, USA) and annealed oligos cloned into pLKO.1 TRC cloning vector (Addgene #10878) using the unique Age1/EcoR1 sites.

A pool of two shRNA plasmids per gene was then packaged into lentiviral particles using psPAX2 and pMD2.G, as previously described [9]. Alternatively, and where indicated, shRNA plasmids (pool of three shRNA plasmids per target) were obtained directly from Santa Cruz.

#### 4.3.3. Lentiviral CRISPR Vectors

The SpCas9 and guide RNA (gRNA) CRISPR components were both expressed from the one-vector lentiviral system “lentiCRISPR_v2” [73] (Addgene #52961). gRNA sequences for target genes were designed by submitting the sequence of an early exon (common to all isoforms) into the CRISPOR prediction tool [74], so that frameshifts in this region would result in at least 70% loss of native protein sequence. gRNAs were selected to meet the tool’s specificity-score requirements and to have at least 4 base pair mismatches with any other exon in the human genome (Appendix A). gRNA oligos were ordered from IDT-Technologies and cloned into the lentiviral vector as outlined by Zhang and colleagues [73] (http://genome-engineering.org/gecko/wp-content/uploads/2013/12/lentiCRISPRv2-and-lentiGuide-oligo-cloning-protocol.pdf, last accessed on the 1 September 2021).

### 4.4. Cell Culture, Genetic Modification and Infection:

Monocytic cell line U937 (ATCC^®^ CRL1593.2™), cultured in RPMI (Thermo Fisher Scientific, Waltham, MA, USA) and supplemented with 10% FCS (Invitrogen) was used in all experiments, unless otherwise stated. The identity of the U937 cell line used throughout this study was verified by microsatellite profile analysis by a NATA accredited third party institution (Garvan Institute, Sydney, Australia) and was confirmed to match the U937 cell line in the ATCC and SMZ databases. The cells were maintained at a density of around 0.5–1 × 10^6^ per ml and passaged every 3 to 4 days. Although HEK F293T and TZMbl cells lines were not verified by microsatellite profile analysis, unique cellular resistance and phenotypic properties of cells were used to ensure purities were maintained. For HEK F239T, this was G418 resistance and the ability to produce HIV viral particles post transfection with the vectors used herein. For the TZMbl line, the ability to be infected by both CCR5 and CXCR4 tropic HIV strains in addition to their ability to produce luciferase and/or ß-galactosidase post infection was used to ensure purities were maintained. All cell lines used in this study were tested to be free of mycoplasma using Mycoalert (Lonza, Basel, Switzerland).

#### 4.4.1. shRNA Depletion

For shRNA knockdown in U937 cell lines, cells were transduced with lentivirus stocks expressing respective shRNA at an MOI of 1.0 and transduced cells selected by passaging in media containing puromycin (2 µg/mL) (Invitrogen) or hygromycin (400 µg/mL) (Invitrogen) for 2 weeks. For double knock down cells, transduced cells stably expressing single gene knockdowns were infected with lentivirus carrying the shRNA for the second target and transduced cells selected by addition of both puromycin and hygromycin to culture media at the above-mentioned concentrations.

#### 4.4.2. CRISPR Gene Knock Outs

For gene-editing using CRISPR vectors, U937 cells were transduced at a MOI of 0.5, and selected in puromycin-containing medium (2 µg/mL) for 7 days. Details of guide RNA used in this study are listed in Appendix A.

#### 4.4.3. Clonal Sorting, Depletion and Knock out Confirmation

Clonal populations for the above-mentioned transduced cell lines were generated by single cell sorting using ARIA flow cytometer (BD Biosciences, Franklin Lakes, NJ, USA) into complete RPMI media that was pre-conditioned for 24 h with the U937 cell line cultured at 2 × 10^5^ cells per mL. Approximately 30% of single cell colonies survived and were grown to confluency. Initially, five clones were selected for live imaging and F-Actin phalloidin staining to reveal any readily observable changes in cortical F-Actin architecture (e.g., Loss of filopodia). Clones were then verified for shRNA depletion and gene knockdown at the protein level was verified by Western blotting of cell lysates using the following antibodies: anti-Arp2 rabbit, polyclonal (Santa Cruz Biotechnology, Dallas, TX, USA; SC-15389, Lot L-2109), anti-Arp3 mouse, monoclonal (Abcam; AB49671, Lot GR450638), anti- Diaph1 rabbit, polyclonal (Bethyl laboratories; A300-077A; Lot A300-077A-2), anti-Diaph2 rabbit, polyclonal (Bethyl laboratories; A300-079A; Lot A300-079A-2), anti-Diaph3 rabbit, polyclonal (Protein tech; 14342-1-AP; Lot 5399), Anti-Rac1 monoclonal (Protein Tech Manchester, UK; 66122-I-Ig; Lot 10011346) and anti-IQGAP1 monoclonal (Santa Cruz Biotechnology, Dallas, TX, USA; sc-376021, Lot H1417) antibodies. In the absence of working antibodies to verify protein target depletion, RT-Q-PCR was performed on FMNL-1 (HS00979762_m1) using Applied Biosystems TaqMan Gene Expression assays; fold change depletion in the shRNA depleted cells was determined by comparison to scrambled shRNA transduced cell controls. In Appendix A, raw data (immunoblotting of cellular lysates) is presented on the clonal cells primarily used in this study. Where possible, CRISPR/Cas9 was used to confirm shRNA depletion phenotypes and to generate cells where the target protein expression was knocked out. Successful gene editing in clonally expanded populations was confirmed by genomic PCR and DNA sequencing with a pair of target-specific “surveyor” primers designed to amplify the gene region targeted by the gRNA (all surveyor primers used are listed in Appendix A). Genomic DNA extraction from CRISPR-edited cells was performed as described in [75]. Briefly, 1 × 10^6^ cells were pelleted, the cell pellet resuspended in 100 µL QuickExtract solution (Lucigen, Middleton, WI, USA.; #QE09050), transferred to PCR-tubes and incubated in a thermocycler for 15 min at 68 °C, followed by 8 min at 95 °C and then held at 4 °C until further use. Of this cell-lysate solution, 2 µL were used as template for genomic PCRs in a 50 µL reaction with Velocity DNA Polymerase (Bioline#21098) following the manufacturer’s instructions. DNA Sanger sequencing was performed by a NATA-accredited core facility (Garvan Institute, Sydney, Australia). Sequencing data was analysed using Sequence Scanner Software v2.0 (Applied Biosystems, Waltham, MA, USA). Note that, in the context of this work, the term knockout (k/o) refers to cellular clones where both gene copies have been edited in a way that results in loss of native protein expression due to introduction of frameshifts or premature stop codons in the protein coding sequence. For representative sequencing data of Cdc42^k/o^ and Wave2 ^k/o^ cell clones, see Appendix A.

Quantitative cell to cell virus transfer assays: Viral transfer assays primarily utilised various U937 clones with shRNA depletion or gene knock outs where indicated. Donor cells were infected at an MOI of 1.0 with VSVg pseudotyped pHIVNL43IRESeGFP by spinoculating at 1200× *g* for 1 h at 18 °C. 48 h post infection, the proportion of infected cells was enumerated by measuring GFP expression via flow cytometry.

Unless otherwise stated, viral transfer assays used the HIV permissive HeLa cell line TZM-bl cells.

5000 infected donor cells normalised to 40% infection were added to 20,000 TZM-bl cells (labelled with “Thermofisher-C34565: CellTracker Deep Red dye” and seeded on the previous day) in a flat bottom 96-well plate (donor to target ratio = 0.25, final well volume = 200 µL). Viral transfer was stopped after 18 h coculture by adding the gp120/ CD4 inhibitor BMS-378806 (Selekchem #S2632) to a final concentration of 10 µm. At 48 h coculture, donor cells were removed by washing twice in PBS. In all cases, the number of infected recipient cells (GFP and Alexa647 double positive, single cells) were enumerated by fluorescence microscopy using the high content CYTELL imaging platform (GE Healthcare). An average of 25,000 cells were analysed per condition by acquiring multiple fields of view.

In virus transfer assays that used primary CD4+ T cells as targets, primary CD4+ T cells were isolated from whole blood. Healthy donors were consented under St Vincent’s Hospital ethics #HREC/13/SVH/145. Briefly, blood from each consenting donor was collected into approximately 10 × 9 mL ACD-B vacuette tubes (Greiner, Cat# 455094, Lot#A180437V). Blood was then pooled and diluted I in 2 with sterile PBS and then subsequent CD4 T cell isolation proceeded as previously described [9]. Following isolation, cells were activated using T Cell TransActTM (Miltenyi Biotech) for 24 h, according to manufacturer’s instructions. CD4+ T cell activation was measured by upregulation of CD69 (Becton Dickinson Clone L78; Lot 41209) and CD25 (Becton Dickinson Clone 2A3; Lot 25912) surface marker expression using flow cytometry. 10^5^ primary activated CD4+ T cells were added per well to a U-bottom 96-well plate in 100 µL of RPMI supplemented with 10% fetal calf serum. Following extensive washing and normalisation of infection to 5%, 5000 donor cells per 50 µL of media were serially diluted at steps of 1 in 5 dilution and added to recipient cells in a final co-culture volume of 150 µL. Four days post co-culture, cells were thoroughly resuspended and added to wells of a 96-well flat bottom tissue culture plate that had been previously coated with poly-L-Lysine as per manufacturer’s instructions (Sigma-Aldrich, St. Louis, MO, USA). As mentioned above GFP positive HIV infected cells were enumerated by fluorescence microscopy using the high content CYTELL imaging platform (GE Healthcare). An average of 25,000 cells were analysed per condition by acquiring multiple fields of view. Under the above conditions, viral transfer is primarily observed to be cell–cell, as supernatants from infected cells cultured alone over this period did not result in significant infection when added to the same target cell types. In addition, the inclusion of the antiretroviral reverse transcriptase inhibitor Efavirenz to co-cultures block eGFP expression within the target cell population (i.e., The GFP signal is not consistent with simple endocytosis in the target cell type).

#### 4.4.4. Quantitative Analysis of Live Cells during Viral Transfer:

Cell–cell contacts

The cumulative number of contacts between each infected donor cell and any uninfected target cells (TZM-bl) was determined via frame-by-frame inspection of 3 h time-lapse movies. Each contact was recorded as a finished track using ImageJ’s MTrackJ plugin [76]. N > 50 cells from two independent experiments were analysed for each group. Cells that exit the field of view were excluded from analysis. Whilst this enumerated absolute number of contacts, cell-to-cell engagement was enumerated when infected donor cells were contacting the same uninfected target cells for a duration greater than 5 min.

Cellular circularity vs. Gag-polarisation

At the virological synapse, donor-cell circularity was calculated using the circularity measurement function in ImageJ (https://imagej.nih.gov/ij/plugins/circularity.html; accessed on the 1 September 2021), where c = 1 indicates a perfect circular shape. Gag polarization was calculated as (IntP/IntD)-1, where IntP is the GFP intensity signal at the proximal cell quarter (i.e., contacting the target cell), whereas IntD is the intensity at the distal cell quarter. A polarization value of 1 indicates 100% enrichment in the proximal quarter.

Immunofluorescence microscopy:

For live and fixed cell imaging, 0.5 × 10^6^ cells were infected with HIV stocks at an MOI of 0.1 for 48 h under standard culture conditions before analysis.

Live cell imaging was performed as described previously using an inverted Olympus IX-70 microscope with 60 × 1.42 NA oil immersion lens and an Evolve 512 back-thinned EM-CCD camera (512 × 512) (DeltaVision ELITE Image Restoration Microscope, GE Healthcare). For time-lapse movies, approximately 5 × 10^4^ cells were seeded onto a 35 mm imaging dish with ibidi polymer coverslip bottom (#1.5) (Ibidi, Martinsried, Germany) and eGFP and DIC channels imaged at approximately 2 frames/sec, with time lapse movies presented as overlays. Manual single particle tracking was done using ImageJ with MTrackJ plugin and filopodial lengths calculated as described previously [9].

For counting virus particles, wells of a 96-well Sensoplate, glass bottom, black (Greiner Bio-One International, Kremsmünster, Austria), were coated with poly-L-lysine, according to the manufacturer’s instructions (Sigma-Aldrich). 50 µL of virus supernatant was serially diluted at 1 in 5 dilutions steps and then added to each well and plate spun at 2500× *g* for 40 min at RT followed by fixation with 4% formaldehyde (*v*/*v*) for 20 min at room temperature. Fluorescent virus particles were then imaged using DeltaVision Elite microscope described above and a total of five fields of view per well were acquired in the GFP channel. Viral particles were then enumerated using ImageJ using the 2D/3D particle tracker in MosaicSuite (MOSAIC Group, Dresden, Germany).

For fixed cell imaging, 5 × 10^5^ cells were cytospun onto 22 × 60 mm #1.5 coverslips (VWR international, Batavia, IL, USA), pre-coated with CellTak (Corning), and then fixed in 4% formaldehyde solution (*v*/*v*) for 20 min at room temperature followed by neutralization with 50 mM NH4Cl (Sigma) for 3 min. Cells were permeabilized with 0.05% Triton-X (Sigma) for 1 min at room temperature, stained with indicated mAbs in the presence of 5% serum, followed by the appropriate secondary antibody. For staining filamentous actin (F-actin), cells were stained with directly conjugated Alexa Fluor-647 Phalloidin (Thermo Fisher Scientific) for 20 min at room temperature prior to mounting in Prolong Gold antifade mountant with DAPI (Thermo Fisher Scientific, Waltham, MA, USA). Cells were visualised with a 100 × 1.4 NA oil immersion lens using a DeltaVision Elite microscope and a Photometrics CoolSnap QE camera. Images were acquired as 50 to 60 optical sections, 0.15 µm to 0.20 µm in thickness, deconvoluted and volume projections generated using DeltaVision SoftWorx software, version 7.0. Applied Precision, Issaquah, WA, USA). Unless otherwise stated, all fixed cell data presented herein are Z-series volume projections.

Correlative FIB-SEM:

In order to perform Correlative FIB-SEM, 0.5 × 10^6^ U937 cells were infected with HIViGFP-Env at an MOI of 0.1. 48 h post infection, cells were washed twice, resuspended in warm RF-10 and incubated at 37 °C, 5% CO_2_ for a further 30 min. This was done to allow the cells to recover the filopodial activity post washing steps. 100μL of cell suspension was then added to a gridded 35 mm glass bottom dish (# 2) from MatTek Corporation (Ashland, MA, USA), that had been pre-coated with CellTak according to the manufacturer’s instructions. Cells were cytospun at 80× *g* for 1 min before fixing in fresh 4% (*v*/*v*) formaldehyde. Cells were then imaged using a DeltaVision Elite microscope. Infected cells were identified by GFP expression and the grid coordinates noted. Cells grown and imaged on gridded coverslips as described above were further fixed in Karnovsky’s fixative prior to transport. The samples were then post-fixed, stained and resin embedded as previously described [20].

Data collection was performed on a Zeiss Crossbeam 540 FIB-SEM (Zeiss Inc., Jena, Germany) controlled by ATLAS3D (Fibics Inc., Ottawa, ON, Canada), as follows. Since the cells were adhered, after the cover slips were removed, the outlines of the cell membranes were just under the top resin surface and could be visualized using a scanning electron beam operated at 3 kV. The pattern of the gridded coverslip was also transferred to the resin surface, allowing correlation with the light microscopic images and location of specific cells of interest. These cells were protected by depositing a layer of platinum on top of the resin, effected by the FIB operated at 700 pA and 30 kV. Notches were FIB milled (at 50 pA) into the platinum to monitor and adjust for specimen drift x, y and z directions as well as beam tuning in real time during data acquisition. Image contrast was achieved by FIB-depositing (at 700 pA) a layer of carbon atop the platinum, and the cells were exposed from the side by FIB milling a trench in the resin at 30 nA followed by 3 nA until the plasma membrane was reached.

During data collection, the FIB and SEM were operated simultaneously, at 30 kV, 700 pA and 1.5 kV, 1 nA, respectively. The EsB detector was operated at a 900 V grid voltage to produce a 2-D stack of images composed of predominantly back-scatter electronic signal. The SEM pixel size was set at 3 or 4 nm and total dwell time of 4 µs, and FIB step size set at 9 or 12 nm. The resulting stack of images were aligned, binned, denoised and inverted with IMOD-based scripts [77] to yield isotropic 3-D image volumes. Segmentation of these reconstructions was completed using Amira (Thermo Fisher Scientific, Waltham, MA, USA) or 3DSlicer [78]. Sub-volumes of veils and filopodia were extracted from these data and fiducials were placed at the locations of budding virions, which could be easily observed. Thus, the number and location of individual virions, as well as the surface area of veils and filopodia, could be quantified using available modules in the XimagePAQ extension for Amira. This extension was also used to calculate mean curvature of veils and filopodia; areas with negative curvatures were named “base” positive curvature as “ridge” or “tip” and neutral curvature as “face” or “shaft”, respectively. Representative features were false colored on a spectrum corresponding to negative (red) to positive (blue) mean curvatures.

Mass spectrometry of purified virions:

Parental and mutant U937 cell lines were infected with HIViGFP^ENV-ve^ at an MOI of 0.3 and after overnight culture cells were washed three time in fresh media to remove the residual inocula. Culture supernatants were then pooled after a harvest at 48 h and 96 h post infections. Virion preparations were depleted with antiCD45 paramagnetic microbeads (Miltenyi Biotech, Gladbach, Germany; cat# 130-045-801), as described previously [32], with the following modifications: Clarified cell culture supernatants were incubated with continuous mixing for 1 h at room temperature with anti-CD45 microbeads at a concentration of 4 µL beads per ml of supernatant. CD45 immunoaffinity depleted supernatants were then separated from beads by placing in magnetic separators, followed by centrifugation at 28,000× *g* for 90 min at 4 °C to pellet virions. Pelleted virions were resuspended in PBS and lysed using 4 × LDS sample buffer (BOLT, Invitrogen). CD45 microvesicle depletion was determined by Western blotting, as previously described [32]. Samples were subsequently separated by SDS-polyacrylamide gel electrophoresis using 1 mm thick 4–12% Bis-Tris gel (Invitrogen), followed by staining with Instant Blue Coomassie stain (Expedeon Inc, San Diego, CA, USA). Each virion containing lane was divided into 18 contiguous sections and each section then subjected individually to in-gel digestion protocol [79]. Briefly, gel slices were de-stained (50% (*v*/*v*) acetonitrile, 50 mM NH4HCO3), reduced (10 mM DTT in 50 mM NH4HCO3) and alkylated (55 mM Iodoacetamide in 50 mM NH4HCO3) before being trypsinised with 100 ng of Trypsin (Promega) for 16 h at 37 °C. Gel slices were then treated with the following solutions sequentially for 1 hour each at RT: 50% (*v*/*v*) acetonitrile/0.1% (*v*/*v*) formic acid and 100% acetonitrile. Samples were then dried in a centrifugal concentrator before resuspending in 20 µL of 0.1% (*v*/*v*) formic acid.

Mass spectrometry

Proteolytic peptide samples were separated by nano-LC using an UltiMate 3000 HPLC and autosampler system (Dionex, Amsterdam, Netherlands) and eluting peptides ionized using positive ion mode nano-ESI following experimental procedures described previously [80]. MS and MS/MS were performed using a Q Exactive Plus mass spectrometer (Thermo Scientific, Bremen, Germany). Survey scans *m*/*z* 300–1750 (MS AGC = 3 × 106) were recorded in the Orbitrap (resolution = 70,000 at *m*/*z* 200). The instrument was set to operate in DDA mode; up to 12 of the most abundant ions with charge states of >+2 were sequentially isolated and fragmented via HCD using the following parameters: normalized energy 30, resolution = 17,500, maximum injection time = 125 ms, and MSn AGC = 1 × 105. Dynamic exclusion was enabled (exclusion duration = 30 s).

Sequence database searches and protein quantification

Sequence database searches were performed using MaxQuant (version 1.5.8.0; Max Planck Institute, Munich, Germany), run using standard parameters using the Andromeda sequence database search utility. Andromeda was employed using the following parameters: precursor ion and peptide fragment mass tolerances were ±4.5 ppm and ±0.5 Da, respectively; carbamidomethyl (C) was included as a fixed modification; oxidation (M) and N-terminal protein acetylation were included as variable modifications; enzyme specificity was trypsin with up to two missed cleavages; human sequences in the Swiss-Prot database (February 2017 release; 20,168 human sequence entries) were searched. Searches were performed with the “match between runs” feature selected, and proteins identified in each gel lane were quantified using the MaxLFQ algorithm employed using standard parameters. A list of selected proteins detected in virion preparations is presented in Appendix A and reductions across different viral preparations are presented in Appendix A.

Statistical tests

OriginPro (Version 9.0; Originlab Corporation, Northhampton, MA, USA) was used to perform statistical analyses and to generate graphs, unless otherwise specified. Unless otherwise stated, the data from two groups were compared; normal distributions were tested using a Shapiro–Wilk test; for normally distributed data, the probability that the mean of each group was significantly different was evaluated using an unpaired Student’s *t* test. For data that is not normally distributed, the probabilities were determined using the Mann–Whitney *U* test. Unless otherwise stated, statistics are summarised in figures as * *p* < 0.01, ** *p* < 0.001 & *** *p* < 0.0001. All data presented herein is representative of a minimum of 3 independent replicates. Where appropriate, the distribution of raw data was presented; no outlying datasets have been excluded for statistical analysis.

## 5. Conclusions

Overall, we propose that HIV has evolved to highjack a specific node of F-actin regulation that positions outgoing virus at the leading edge of cortical F-actin structures. Since filopodia play an important role in scanning of the microenvironment and mediating immune cell interactions, their corruption is beneficial to the virus, because it biases the first line of cell–cell contacts towards HIV spread, e.g., by providing enhanced adhesion and specificity to CD4+ target cells. However, as these contacts form and the vs. matures, the relationship of HIV with F-Actin changes; observations herein support this change to facilitate final viral release. We conclude that manipulation of the Cdc42-IQGAP1-Arp2/3 actin regulatory node is essential for corruption of the cells’ leading edge to facilitate cell–cell HIV spread; observations support a key role for IQGAP1 during later cell–cell contact as well. Moving forward, greater spatiotemporal resolution of this latter events is needed and will give further insight into why many pathogens such as HIV have evolved to interact with IQGAP1 and its binding partners.

## Figures and Tables

**Figure 1 pathogens-11-00056-f001:**
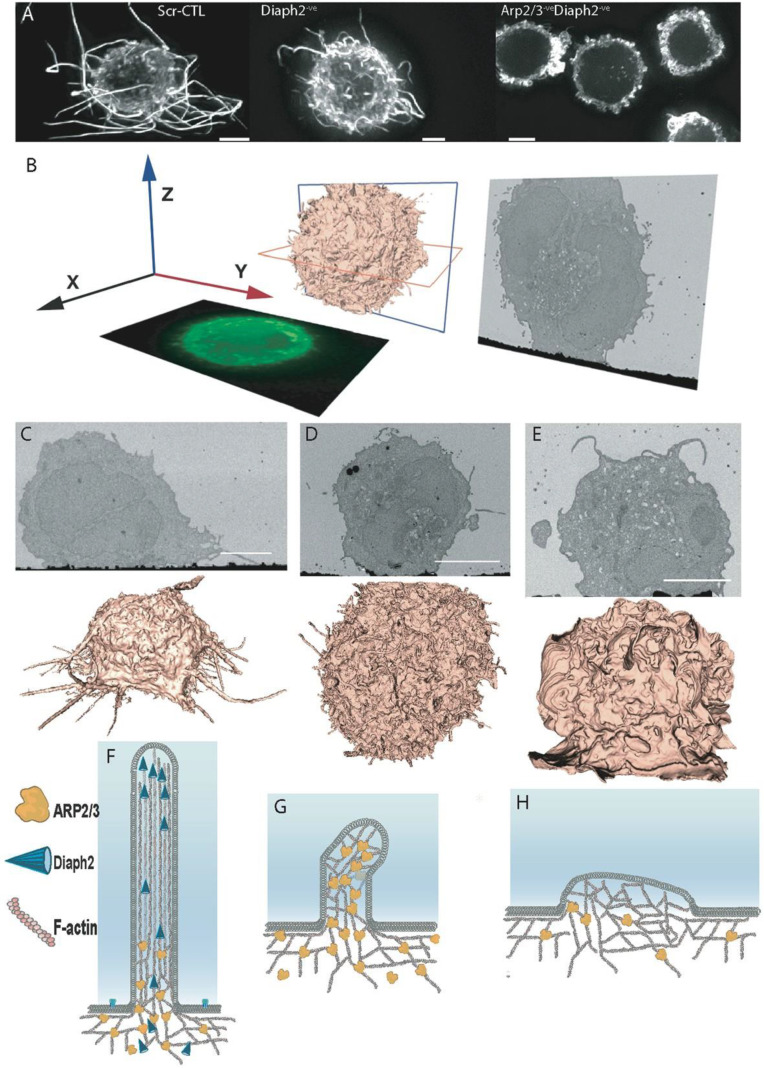
Depletion of F-actin regulators reveals long filopodial networks to be driven by Arp2/3 and Diaph2. (**A**) F-Actin staining by phalloidin Alexa-647 reveals extensive and long filopodia in uninfected scramble shRNA control cells (Scr-CTL), short abundant filopodia upon deletion of the formin Diaph2, and extensive lamellipodia when both Diaph2 and Arp2/3 are co-depleted. (**B**) Schematic representation of FIB-SEM imaging data. (**C**–**E**) FIB-SEM 2D images (upper) and 3D reconstructions (lower) of (**C**) non-depleted control cells (Scr-CTL), (**D**) Diaph2-depleted cells and (**E**) Diaph2 and Arp2/3 co-depleted cells. All scale bars are 5 μm. (**F**–**H**) Schematic representation of phenotypes induced by loss of F-Actin regulators. (**F**) Wildtype scenario, (**G**) Diaph2 deficiency, (**H**) Diaph2 & Arp2/3 deficiency. Note in (**H**) we represent reduced levels of Arp2/3, given its high cellular abundance and residual levels of Arp2/3 in our knockdown cells (Appendix A). From here on, in all figure legends, all cells used to observed the dynamics of F-Actin during HIV infection are based on the U937 cell line or related genetically manipulated clones.

**Figure 2 pathogens-11-00056-f002:**
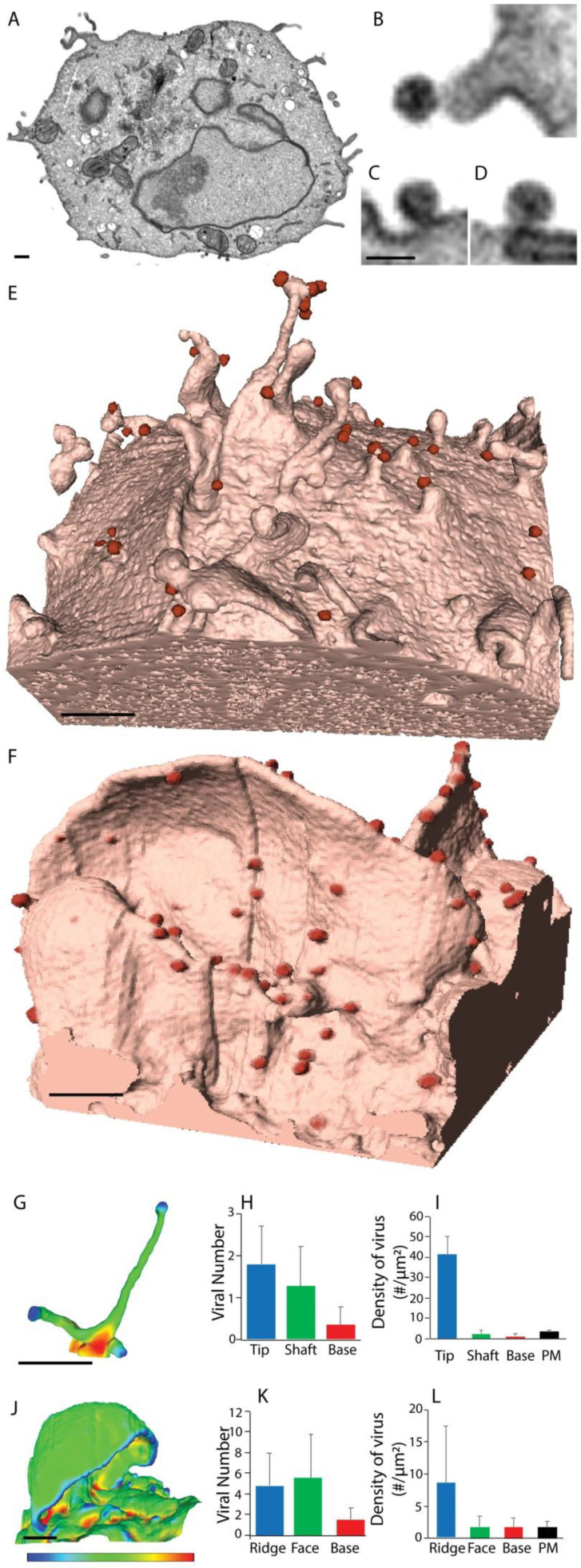
HIV budding enriched to positively curved cortical F-Actin. (**A**) Representative FIB-SEM images of HIV virions associated with (**B**) Filopodia and (**C**,**D**) plasma membrane of a HIV infected cell depleted of Diaph2. (**E**,**F**) 3D rendering FIB-SEM images of HIV infected (**E**) Diaph2^−ve^ and (**F**) Diaph2^−ve^Arp2/3^−ve^ cell clones. HIV buds are shaded in red to highlight their location. (**G**,**J**) enumeration of HIV-buds in association with positively curved cortical F-Actin structures. (**G**) filopodia and (**J**) lamellipodia are pseudocolored using a colour spectrum from blue (positive curvature), green (neutral curvature) to red (negative curvature). (**H**,**I**) Enumeration of total HIV buds in association with the filopodia in Diaph2^−ve^ cells. (**H**) Absolute viral bud counts and I. HIV bud density per μm^2^. (**K**,**L**) Enumeration of total HIV buds in association with the lamellipodia in Diaph2^−ve^ Arp2/3^−ve^ cells. (**K**) Absolute viral bud counts and (**L**) HIV bud density per μm^2^. All scale bars are 0.5 μm, with the exception of (**B**–**D**), which is 0.1 μm. Bar graphs in (**H**,**I**) represent mean and standard deviations of virion counts from *n* = 15 filopodial structures. (**K**,**L**) bar graphs represent mean and standard deviations of virion counts from *n* = 15 lamellipodial structures. Statistics for panels (**G**–**L**) are summarised in Appendix A.

**Figure 3 pathogens-11-00056-f003:**
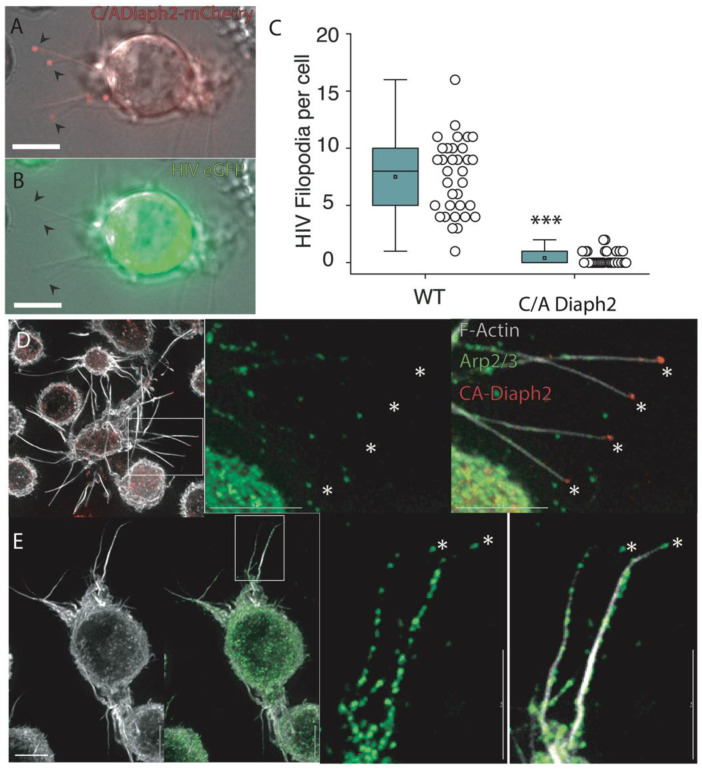
Constitutively active Diaph2 driven Filopodia are not associated with HIV buds. (**A**,**B**) Live still images from Appendix A; (**A**) ^C/A^Diaph2-mCherry (red) positive cells, infected with (**B**) HIViGFP (green). Note Diaph2 at the tips of filopodia are negative for HIV. Scale bars are at 5 μm. (**C**) Quantification of HIV positive filopodia per cell in HIV infected live cell cultures. HIV filopodia counts are derived from three independent HIV infections *** *p* < 0.0001. (**D**,**E**) Immunofluorescent Arp2/3 staining of (**D**) ^C/A^Diaph2-mCherry (red) positive cells and (**E**) Untreated cells. Asterisks (*) highlight the terminal ends of filopodia that are either Diaph2 positive & Arp2/3 negative (for ^C/A^Diaph2-mCherry) or Arp2/3 positive (for untreated cells).

**Figure 4 pathogens-11-00056-f004:**
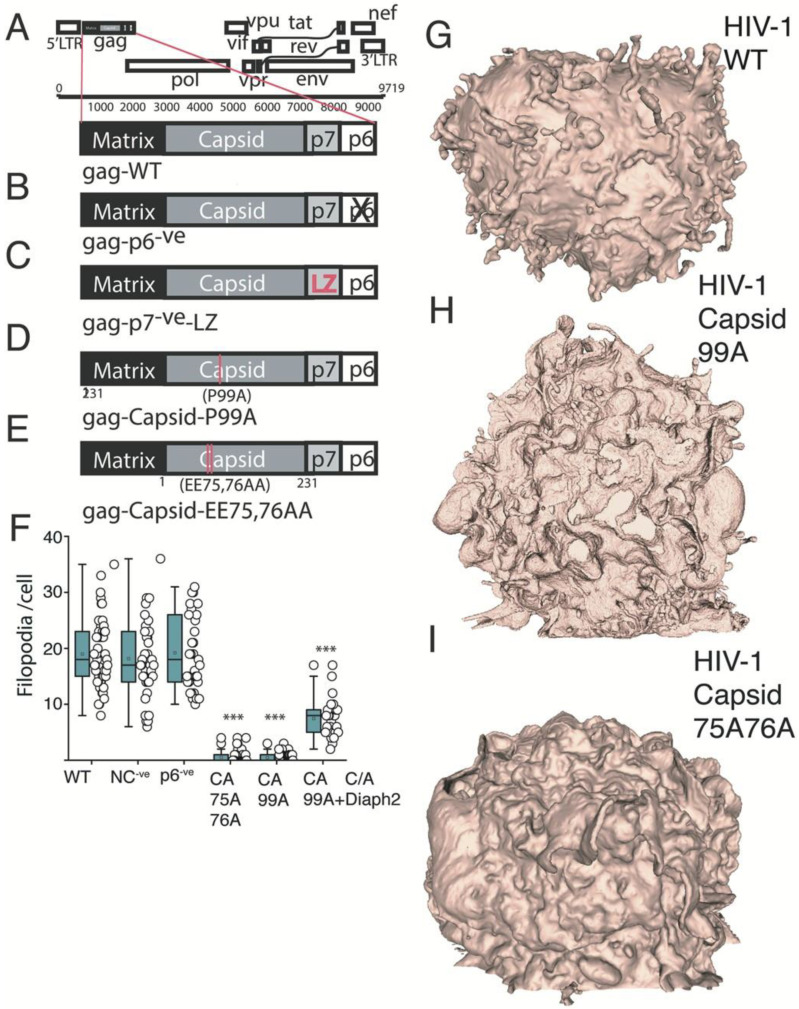
HIV Gag curvature mutants can impact Arp2/3 dependent cortical F-Actin (**A**–**E**) HIV Gag mutants used. (**A**) Wild type HIV Gag. (**B**) HIV Gag late domain mutant with p6 deleted. (**C**) NC deletion mutant with the Leucine Zipper (LZ) derived from *Saccharomyces cerevisiae* GCN4 to rescue Gag oligomerisation. (**D**,**E**) HIV Gag curvature mutants (**D**) P_99_A and (**E**) EE_75_,_76_AA. (**F**) Enumeration of filopodia per cell in Diaph2^−ve^ cell clones. *** *p* < 0.0001. (**G**,**H**) Representative FIB-SEM 3D rendered images of (**G**) WT vs. curvature mutants (**H**) P_99_A and (**I**) EE_75,76_AA. Data is derived from three independent HIV infections using each mutant.

**Figure 5 pathogens-11-00056-f005:**
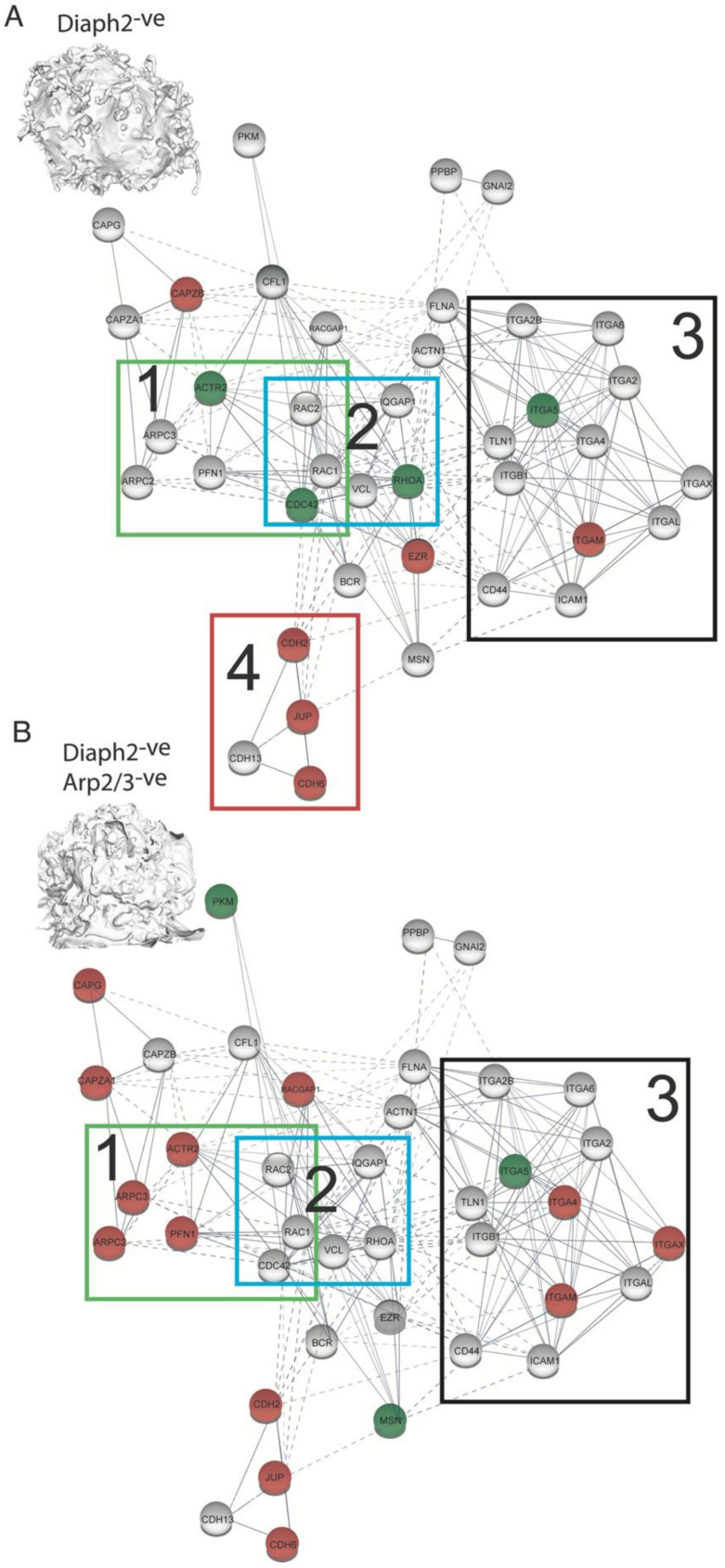
Cortical F-Actin regulators enriched at the final stages of HIV egress are revealed through HIV proteomics. (**A**,**B**) Viral proteome analysis of proteins associated with cortical F-Actin regulation. Proteins with increased abundance relative to untreated cells are shown in green, whereas those with relative decreased abundance are shown in red. (**A**) Virions produced in Diaph2^−ve^ cells. **1**. Highlights the Arp2/3 complex node where the amounts of ACTR2 and Cdc42 are increased in virion proteomes. **2**. Indicates a GTPase node in association with IQGAP1. **3.** Highlights a node of integrin and related proteins. **4**. Highlights a node involved in cadherin adhesion that is downregulated upon Diaph2 depletion. (**B**) Virions produced in Diaph2^−ve^Arp2/3^−ve^ cells. Note that in 1. Arp2/3 components are predictably depleted compared to untreated cells, whilst in 2. the GTPase node and IQGAP1 remain unchanged.

**Figure 6 pathogens-11-00056-f006:**
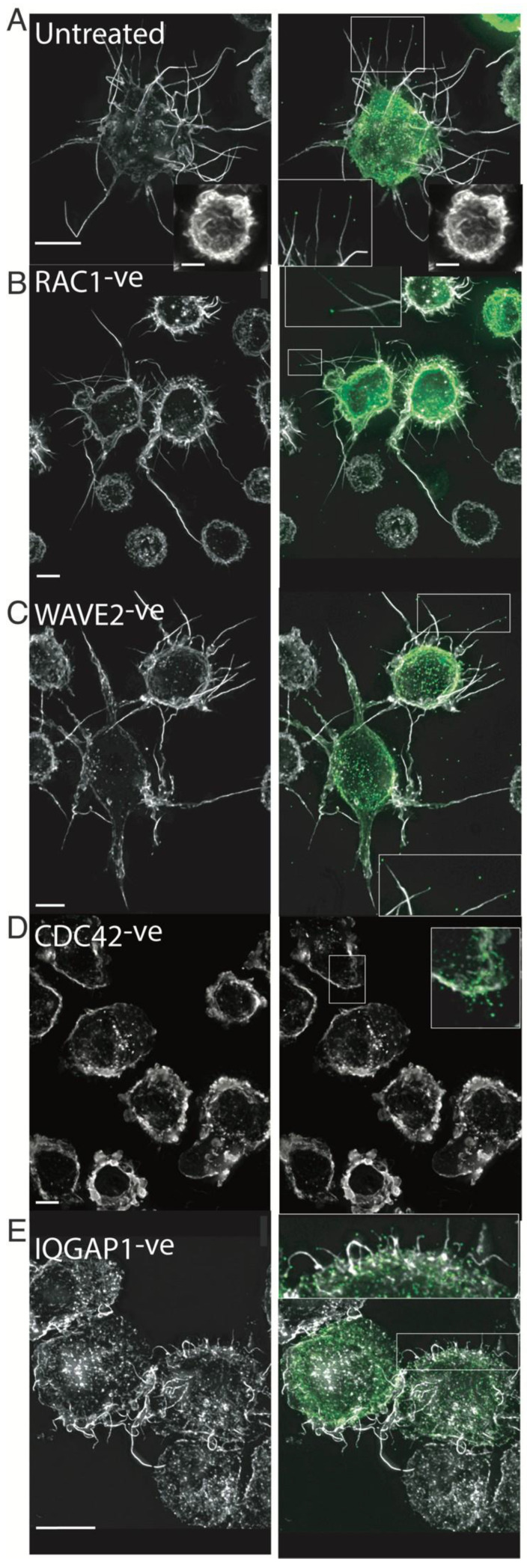
HIV infection and its influence on cortical F-Actin. (**A**) From viral proteomics, we identified a common node of actin regulators associated with Arp2/3-dependent filopodia and lamellipodia. Through shRNA depletion or CRISPR-Cas9 knockout, we generated clonal cell populations depleted of various actin regulators. (**B**,**C**) Lamellipodial regulators. (**B**) Rac1^−ve^ and (**C**) WAVE2^k/o^. (**D**) Cdc42^k/o^ (filopodial regulator). (**E**) IQGAP1^−ve^. (**A**) Represents the untreated infected control. The image inset in (**A**) is an uninfected untreated control for comparison. All cells were infected with HIV iGFP (green) and then counterstained with phalloidin Alexa-647 (white). All scale bars are at 5 μm. Inset magnifications reveal HIV at the tips of filopodial structures. Note in Rac1^−ve^ and WAVE2^k/o^ images the extensive filopodial networks only present in HIV infected (green) cells (see Appendix A for formal quantification).

**Figure 7 pathogens-11-00056-f007:**
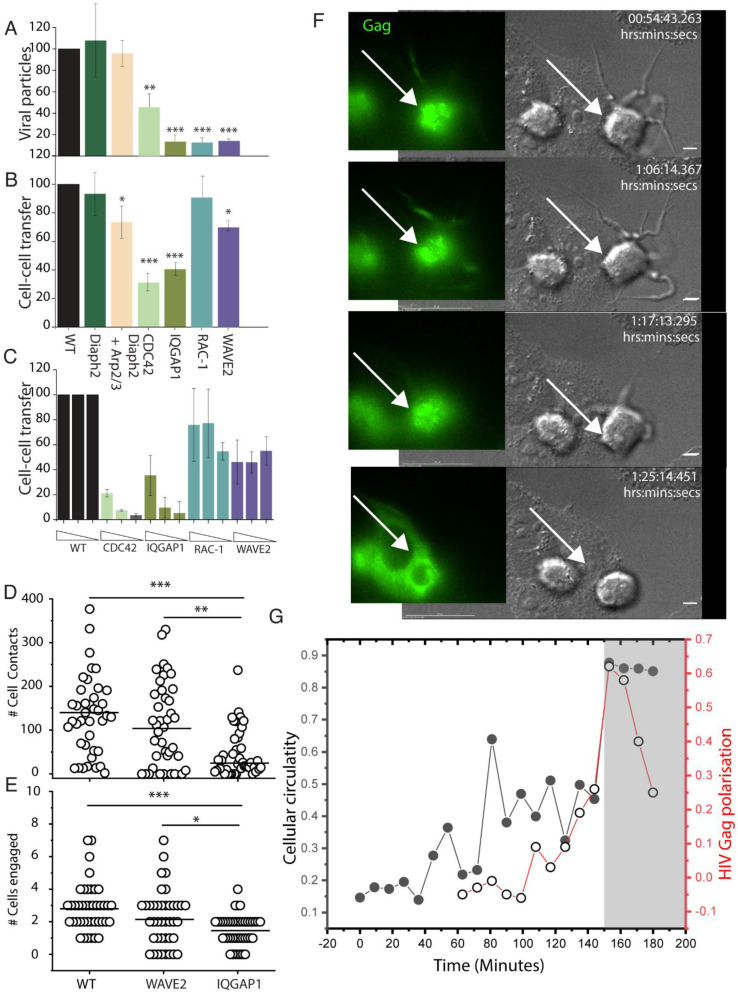
HIV spread is dependent on Cdc42 and IQGAP1 (**A**) All cells are infected with pHIVNL43iGFP and normalised to 5% infection on day 3. After normalisation, cell supernatants are collected over a 24 h period and GFP positive HIV particles are spinoculated onto 96-well glass plates coated with poly-L-lysine. Absolute viral particle counts are determined by high- resolution fluoresecence microscopy per 4 fields of view. Herein, the data is presented as a relative count ((virion count/virion count in WT control) × 100). (**B**) HIV infected cells as normalised in (**A**) are then co-cultured at a ratio of 1:5 with HIV-permissive TZMBl targets. (**C**) Infected cells are co-cultured with primary CD4 target T-cells at limiting dilutions. Dilution steps correspond to 5%, 1% and 0.2% infected cells in the donor population. Exposure to virus from infected cells is limited to 24 h, after which an entry inhibitor BMS806 is added to prevent further viral spread. (**A**–**C**) Data indicates the mean and standard deviation from 3 independent experiments. In (**C**), primary recipient CD4 T cells were sourced from independent blood donors. (**D**) The cumulative number of contacts between each infected donor cell and any uninfected target cells (TZM-bl) over 3 h. (**E**) same as (**D**), but only the first contact with each distinct target cell is counted. (**F**) Representative example of a time-lapse series from (**D**,**E**). Cells are infected with HIViGFP, allowing real-time visualization of Gag. (**G**) At the virological synapse, donor-cell circularity was used to enumerate lack of membrane protrusions (i.e., Lack of filopodia) in parallel with Gag polarization. In grey shading are the time points where GFP cytoplasmic transfer (post-synaptic donor-target cell fusion) is observed. * *p* < 0.01, ** *p* < 0.001, *** *p* < 0.0001.

## Data Availability

Due to the large size of files datasets generated in Figure 1, Figure 2, Figure 4, Figure 5 and Figure 7, the data will be available using the Dryad Data Repository.

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
