# Peer review of "Embedding of HIV Egress within Cortical F-Actin"

_pathogens, 2022, doi:10.3390/pathogens11010056_

Round 1

Reviewer 1 Report

In this manuscript, Aggarwal et al investigated how HIV utilizes cortical F-actin cytoskeleton for highly efficient cell-cell viral spread.  They discovered that HIV exploits a Cdc42-IQGAP1-Arp2/3 filopodial pathway to facilitate embedding of HIV egress at the leading edge of cortical F-actin.

The overall study is well controlled, and the conclusions are supported by a large body of evidence based on newly generated cell lines, carefully adapted advanced imaging techniques, etc.  Importantly, the findings are largely interpreted with caution and the discussions are comprehensive and insightful.  I therefore only have a few minor suggestions for improvement.

Major:

I understand that this study was focused on F-actin and hence only proteins associated with cortical F-Actin regulation were reported in the HIV proteomic experiment (Fig. 5). Yet I am curious if other proteins of significance were identified in the proteomic analysis. For instance, in an earlier study, Popov et al reported that HIV-1 gag recruits a host protein called PACSIN2 to promote virus cell-cell spreading (Popov S, et al, Proc Natl Acad Sci USA, 2018, 115:7093–7098.).

Minor:

Line 33, remove “to”.

Line 129, “Fig.1 E&H” should be “Fig.1 D&G”.

Line 333, remove “the address”.

Line 335, “resolved” should be “resolve”.

Line 442, remove “needs to”.

Line 461, 463, 510, remove “as”.

Line 479, remove “is”.

Line 487 and 490, awkward usage of “we support” in the sentence.   

Author Response

Major:

I understand that this study was focused on F-actin and hence only proteins associated with cortical F-Actin regulation were reported in the HIV proteomic experiment (Fig. 5). Yet I am curious if other proteins of significance were identified in the proteomic analysis. For instance, in an earlier study, Popov et al reported that HIV-1 gag recruits a host protein called PACSIN2 to promote virus cell-cell spreading (Popov S, et al, Proc Natl Acad Sci USA, 2018, 115:7093–7098.).

Yes we are aware of this work, as it appeared during our study. We did not observe PACSIN2 in our virion lysates. That said, in other cell types or under conditions not tested in our study, it may indeed appear.

Minor:

Line 33, remove “to”.

Line 129, “Fig.1 E&H” should be “Fig.1 D&G”.

Line 333, remove “the address”.

Line 335, “resolved” should be “resolve”.

Line 442, remove “needs to”.

Line 461, 463, 510, remove “as”.

Line 479, remove “is”.

Line 487 and 490, awkward usage of “we support” in the sentence.   

Minor corrections listed above have been corrected.

Reviewer 2 Report

Aggarwal and colleagues have expanded their previous findings on HIV budding at filopodia and its role in viral spread. In the current study, they use fluorescent live cell imaging and focused ion beam scanning electron microscopy (FIB-SEM) to demonstrate that a Cdc42-IQGAP1-Arp2/3 pathway plays an important role in orchestrating HIV budding and F-actin remodeling in a monocytic cell line. This report also sheds lights on the role of positive membrane curvature in HIV budding. While these findings are interesting and important to understand HIV transmission, several points need to be addressed to reveal the relevance of these findings.

Major concerns:

  1. The primary findings came entirely from experiments using U937 cells and were not validated with primary cells such as dendritic cells (DCs).

  1. It is stated that U937 cells is used as a model of hematopoietic cell lineages (Line 107), which is somewhat misleading and should be rephrased. U937 is a pro-monocytic cell line and has been used as a myeloid cell model. In fact, in the previous study from the same group used U937 cell line as a model of DCs since they have similarity in length and frequency of viral filopodia (PMID: 22685410). The authors should clarify this point as to which cell type(s) U937 cells model for. If U937 cells are used as a DC model in the current study as well, it should be noted that the role of Cdc42 in DC-to-T cell transmission has been already published by Nikolic et al. (PMID: 21562048), which is neither discussed nor cited in the current manuscript.

  1. In Figure 4, the authors tested HIV Gag mutants that are deficient for Gag-induced membrane curvature. Although this is an elegant approach, control experiments are missing to show expression levels of Gag mutants in U937 cells and their localization in particular membrane localization.

  1. TZM-bl cells were used as a recipient cell in some cell-to-cell transmission experiments. In their previous report (PMID: 22685410), the authors’ group has shown that U937-Jurkat cell transmission is Diaph2 dependent, while this study finds U937-TZM-bl cell transmission is Diaph2 independent (Fig 7B). So, it is obvious that a target cell type also affects the requirement of cellular components for cell-to-cell transmission. In Fig 7C, primary CD4 T cells were tested as a recipient cell, but for some reason Diaph2-negative U937 and Diaph2-Arp2/3-double negative U937 cells were not included. These experiments should be repeated with Diaph2-negative U937 and Diaph2-Arp2/3-double negative U937. Live cell imaging of cell-to-cell transmission assay should be validated with primary CD4 T cells or at least with Jurkat cells which are much more relevant than TZM-bl cells.

Minor concerns:

  • In Figure 7, HIV spread from U937 to TZM-bl cells were tested. In the current experimental setup (3-hour incubation), the observed GFP expression in TZM-bl cells were most likely due to cell-to-cell fusion (GFP cytoplasmic transfer). To avoid confusion, cell-to-cell fusion and cell-to-cell transmission should be distinguishably described. The term “Viral fusion” (Fig 7) is inaccurate.

  • U937 cell is not specified in figure legends.

Author Response

Major concerns:

  1. The primary findings came entirely from experiments using U937 cells and were not validated with primary cells such as dendritic cells (DCs).

The initial aim of our study was to map a F-Actin pathway that linked HIV to F-Actin structures we had previously identified in primary cells. Whilst we understand the primary concern here, it was not feasible to carry out the in-depth mechanistic studies using terminally differentiated dendritic cells. In the majority of cases we genetically disrupted genes in this cell line, clonally isolated this cell and then investigated its impact on HIV infection with regards to F-Actin dynamics. For instance, in many cases cells were modified, cloned, verified and then modified a second time, cloned and verified. Whilst technologies like CRISPR-Cas9 gene disruptions are powerful, their use in primary cells are in its infancy & immature dendritic cells would not sustain this type of genetic manipulation without a significant influence on their maturation phenotype. In addition, they cannot be cloned and would represent a continuum of genetically edited and non-edited cells.  To add, many cytoskeletal pathways have been mechanistically dissected in cell-free systems and this study is only one of a few that has mapped a pathway that demonstrated Arp2/3 filopodia seeding that is later elongated with hDiaph2. Indeed, with regard to the latter, there are limited observations of hDiaph2 role in filopodia formation, as it was always assumed that the human equivalent of murine Diaph2, hDiaph3, played this role.

  1. It is stated that U937 cells is used as a model of hematopoietic cell lineages (Line 107), which is somewhat misleading and should be rephrased. U937 is a pro-monocytic cell line and has been used as a myeloid cell model. In fact, in the previous study from the same group used U937 cell line as a model of DCs since they have similarity in length and frequency of viral filopodia 1. The authors should clarify this point as to which cell type(s) U937 cells model for. If U937 cells are used as a DC model in the current study as well, it should be noted that the role of Cdc42 in DC-to-T cell transmission has been already published by Nikolic et al. 2, which is neither discussed nor cited in the current manuscript.

There was no intention to be misleading in our manuscript. We state very clearly what cell line is used in our study.   U937 is a human monocytic cell line of myeloid hematopoietic lineage 3 that is widely recognized as valid experimental model to study the biology and behaviour of monocytes and their myeloid progeny 1,4,5.  Importantly, U937 share the expression pattern of lineage-restricted cytoskeletal regulator factors that are enriched in primary myeloid cells 6,7, and can even be differentiated to macrophages or dendritic cells by different stimuli. Therefore, U937 are a physiologically relevant but versatile and genetically malleable system to study the intersection of HIV and F-Actin in human myeloid cells, including monocytes, macrophages, and dendritic cells.

To clarify this further, we have modified the first paragraph of the results section as follows:

“A physical association of HIV with F-actin structures has been previously observed in all major HIV primary target cell types 1,8. In infected CD4+ T-cells and dendritic cells this manifests in the form of HIV-Filopodia, which are F-actin rich finger-like structures with HIV assembly observed at their tips 1.  Since these structures are more prominent on dendritic cells and similarly enriched in U937 cells  1,9,10, this latter promonocytic cell line provides an ideal model to dissect the link between F-Actin and HIV assembly in the specific context of myeloid hematopoietic cell lineages.”.

With regard to the Nikolic paper, it was not referenced as the mechanisms observed within out work are not the same as that observed in this previous work. For instance there are significant differences in the structures and mechanisms that HIV is interacting with. For instance, Nikolic et al. have previously also reported Cdc42 and Arp2/3 to be critical for the formation of F-actin structures that contribute to cell-cell transfer of HIV from myeloid cells when not productively infected  2.  However, despite the apparent overlap with our findings, it is important to note that these structures are fundamentally different from HIV-filopodia, in terms of their ultrastructure, physical association with HIV, mechanisms of cytoskeletal manipulation involved, and the way in which they promote viral transfer.  Firstly, those structures originally resembled filopodia in transmission electron microscopy images 2, however further ultrastructural analysis later showed that they were in fact thin-walled “sheet-like” projections decorated with fully extracellular virions 11.  For simplicity, we will refer to these structures as “HIV-sheets”.  In contrast, our FIB-SEM data confirms that our structures correspond to what is defined as filopodia (see definition from 12 with immature HIV-buds on their surface that have not yet undergone abscission from the host cell membrane.  Although Cdc42 and Arp2/3 are required for formation of both HIV-sheet and HIV-Filopodia, manipulation of these actin regulators by the virus is fundamentally different because it occurs at opposing sides of the host cell membrane. HIV-sheets form in an Env-dependent manner within minutes of dendritic cells being exposed to extracellular virions 2. Therefore, it is independent of productive infection and does not require de novo expression of viral proteins. Rather, it is believed to be a consequence of Src signalling upon binding of Env to DC-SIGN on the outer cell membrane 13. In contrast, HIV-Filopodia formation is entirely dependent on intracellular expression of immature Gag and is completely independent of the presence of Env or the activation of Src 1. Finally, since HIV-sheets form in uninfected cells they can only mediate cell-cell transfer “in trans”, whereas HIV-Filopodia form in infected cells, which can also mediate viral transfer “in cis”.  Overall, the fact that HIV exploits common actin regulators for facilitating HIV transfer both in cis and in trans highlights how critical this biological host-pathogen intersection is for the virus, and it encourages intervention strategies targeting these common regulators, as this would likely counteract both modes of HIV spread.

To address this point and resolve how our work sits with the contribution of others, we have edited to discussion to highlight the novel observations of our work and clarify how they diverge from this work:

“Whilst others have also reported Cdc42 and Arp2/3 to be critical for the formation of F-actin structures that contribute to cell-cell transfer of HIV from dendritic cells 2, it must be noted there are two distinct differences with respect to these structures. Firstly they are not filopodia but rather sheet-like F-Actin structures 2,11.  Secondly, these F-Actin sheets form in uninfected cells that can mediate cell-cell transfer “in trans. In the study described herein, HIV-Filopodia form in infected cells and participate in cell-cell spread when a donor cell is productively infected (“in cis”).  Overall, the fact that HIV exploits common actin regulators for facilitating HIV transfer both in cis and in trans highlights how critical this biological host-pathogen intersection is for the virus, and it encourages intervention strategies targeting these common regulators, as this would likely counteract both modes of HIV spread.”

  1. In Figure 4, the authors tested HIV Gag mutants that are deficient for Gag-induced membrane curvature. Although this is an elegant approach, control experiments are missing to show expression levels of Gag mutants in U937 cells and their localization in particular membrane localization.

The majority of work herein is correlative FIB-SIM electron microscopy. That is, a cell is identified within a grid using fluorescence & then that same coordinate is mapped and then processed for FIB-SIM. To demonstrate what each HIV Gag mutant looks like when expressed at the fluorescent level we have included representative images of cells infected with these viral mutants as Figure S5.

  1. TZM-bl cells were used as a recipient cell in some cell-to-cell transmission experiments. In their previous report 1, the authors’ group has shown that U937-Jurkat cell transmission is Diaph2 dependent, while this study finds U937-TZM-bl cell transmission is Diaph2 independent (Fig 7B). So, it is obvious that a target cell type also affects the requirement of cellular components for cell-to-cell transmission. In Fig 7C, primary CD4 T cells were tested as a recipient cell, but for some reason Diaph2-negative U937 and Diaph2-Arp2/3-double negative U937 cells were not included. These experiments should be repeated with Diaph2-negative U937 and Diaph2-Arp2/3-double negative U937. Live cell imaging of cell-to-cell transmission assay should be validated with primary CD4 T cells or at least with Jurkat cells which are much more relevant than TZM-bl cells.

For live cell imaging, there are technical complexities involved in imaging two non-adherent cells and thus why we have used a permissive adherent cell to observe the dynamics of F-Actin in our non-adherent donor cells as they engaged the TZM-bl cell line. As the conditions required to extensively live image the synapse in non-adherent cells will take some time, this is beyond the current work that is under review and will be dissected in future studies.  For additional cell-cell transfer experiments involving Diaph2-Arp2/3-double negative U937 cells, see response to reviewer#3 below.

Minor concerns:

In Figure 7, HIV spread from U937 to TZM-bl cells were tested. In the current experimental setup (3-hour incubation), the observed GFP expression in TZM-bl cells were most likely due to cell-to-cell fusion (GFP cytoplasmic transfer). To avoid confusion, cell-to-cell fusion and cell-to-cell transmission should be distinguishably described. The term “Viral fusion” (Fig 7) is inaccurate.

We have removed the term viral fusion, and edited the text as follows to avoid confusion. ”In grey shading are the time points where GFP cytoplasmic transfer (post-synaptic donor-target cell fusion) is observed”.  We have also removed the unnecessary label “viral fusion” from Figure 7.G.

U937 cell is not specified in figure legends.

It is stated in the first section of the results that the model is the U937 cell line. To avoid confusion, we have stated in the first figure legend:

“From hereon, in all figure legends all cells used to observe the dynamics of F-Actin during HIV infection are based on the U937 cell line or related genetically manipulated clones.”

References cited within this cover letter

  1. Aggarwal A, Iemma TL, Shih I, et al. Mobilization of HIV spread by diaphanous 2 dependent filopodia in infected dendritic cells. PLoS pathogens. 2012;8(6):e1002762.
  2. Nikolic DS, Lehmann M, Felts R, et al. HIV-1 activates Cdc42 and induces membrane extensions in immature dendritic cells to facilitate cell-to-cell virus propagation. Blood. 2011;118(18):4841-4852.
  3. Sundstrom C, Nilsson K. Establishment and characterization of a human histiocytic lymphoma cell line (U-937). Int J Cancer. 1976;17(5):565-577.
  4. Liu L, Zubik L, Collins FW, Marko M, Meydani M. The antiatherogenic potential of oat phenolic compounds. Atherosclerosis. 2004;175(1):39-49.
  5. Liu H, Fang S, Wang W, et al. Macrophage-derived MCPIP1 mediates silica-induced pulmonary fibrosis via autophagy. Part Fibre Toxicol. 2016;13(1):55.
  6. Ponten F, Jirstrom K, Uhlen M. The Human Protein Atlas--a tool for pathology. J Pathol. 2008;216(4):387-393.
  7. Wu C, Jin X, Tsueng G, Afrasiabi C, Su AI. BioGPS: building your own mash-up of gene annotations and expression profiles. Nucleic Acids Res. 2016;44(D1):D313-316.
  8. Eugenin EA, Gaskill PJ, Berman JW. Tunneling nanotubes (TNT) are induced by HIV-infection of macrophages: a potential mechanism for intercellular HIV trafficking. Cellular immunology. 2009;254(2):142-148.
  9. Bourinbaiar AS, Phillips DM. Transmission of human immunodeficiency virus from monocytes to epithelia. Journal of acquired immune deficiency syndromes. 1991;4(1):56-63.
  10. Pearce-Pratt R, Malamud D, Phillips DM. Role of the cytoskeleton in cell-to-cell transmission of human immunodeficiency virus. Journal of virology. 1994;68(5):2898-2905.
  11. Felts RL, Narayan K, Estes JD, et al. 3D visualization of HIV transfer at the virological synapse between dendritic cells and T cells. Proceedings of the National Academy of Sciences of the United States of America. 2010;107(30):13336-13341.
  12. Mattila PK, Lappalainen P. Filopodia: molecular architecture and cellular functions. Nature reviews Molecular cell biology. 2008;9(6):446-454.
  13. Shrivastava A, Prasad A, Kuzontkoski PM, Yu J, Groopman JE. Slit2N Inhibits Transmission of HIV-1 from Dendritic Cells to T-cells by Modulating Novel Cytoskeletal Elements. Sci Rep. 2015;5:16833.
  14. Ladinsky MS, Kieffer C, Olson G, et al. Electron tomography of HIV-1 infection in gut-associated lymphoid tissue. PLoS pathogens. 2014;10(1):e1003899.
  15. Sabo Y, de Los Santos K, Goff SP. IQGAP1 Negatively Regulates HIV-1 Gag Trafficking and Virion Production. Cell Rep. 2020;30(12):4065-4081 e4064.
  16. Swart-Mataraza JM, Li Z, Sacks DB. IQGAP1 is a component of Cdc42 signaling to the cytoskeleton. The Journal of biological chemistry. 2002;277(27):24753-24763.

Reviewer 3 Report

General comments:

In this manuscript the authors use shRNA-mediated knockdown and CRISPR/Cas9-mediated knockout of multiple components of the actin cytoskeleton regulatory machinery in U-937 monocytic cells to investigate, via live cell imaging and FIB-SEM, how HIV highjacks the cell’s F-actin remodeling mechanisms to promote its spreading to target cells via filopodia-mediated cell-cell contacts. They found that interference with a Rac1-Wave2-Arp2/3 lamellipodial pathway compromised HIV budding and viral particle release but still allowed cell-cell HIV transfer, whereas interference with a Cdc42-IQGAP1-Arp2/3 filopodial pathway impaired both HIV budding and cell-cell transfer. They propose that by activating the filopodial pathway HIV sequesters the actin remodeling machinery to enable placement of HIV buds at the cell’s leading edge, which initially prevents particle release but, upon maturation of cell-cell contacts, enables viral release and infection of the contacting cell.

The manuscript is very well written and, for the most, experimentally sound. However, there are several points that need to be addressed:

Specific comments:

1) The significance of this study is limited by the fact that all conclusions on the impact of HIV in actin remodeling were withdrawn from experiments with clones isolated from one single immortalized monocytic cell line - U-937. Given that there is recent work that proposes that IQGAP1 signaling negatively regulates HIV trafficking, budding and virion egress, it would be important for the authors to show that their observations are not exclusive to the cell model utilized. It would be valuable if the authors could demonstrate the relevance of Cdc42 and Rac1 signaling for viral egress and cell-cell transfer in primary cells, namely by using chemical inhibitors such as EHT1864 and ML141, or even the Arp2/3 inhibitor CK-666.

2) Still regarding the Arp2/3 complex, since the authors argue the importance of its activation downstream of the HIV-stimulated Cdc42-IQGAP and Rac1-Wave2 pathways, how do they justify that there is no appreciable decrease in viral transfer nor in virion release in Diaph2-ve+Arp2/3-ve cells (re: Fig7)?

3) Still on this note, the discussion should be streamlined and speculative sentences and conclusions avoided. For example, in lines 474-475 – the authors state that “(…) with IQGAP1 recruited by HIV Gag, Cdc42 would be maintained in an active GTP bound state [43]”, however, to my knowledge, there is no evidence, and certainly not in the cited reference, that binding to IQGAP can maintain Cdc42 in an active conformation. To sustain this claim, the authors would have to use CRIB-pulldown assays to demonstrate this locking of Cdc42 in the GTP-bound state after HIV gag-mediated recruitment of IQGAP upon Rac1 depletion.

4) Another example can be seen in lines 483-484: “(…) we hypothesise leading edge structures positive with HIV can indeed indirectly contribute to cell-cell spread and that only a proportion of HIV buds at the membrane engage in this process.” Then, shouldn't this fail in cells deprived of lamellipodia through Rac1/Wave depletion?

In the HIV egress proteomic analysis the abundance of RhoA also increased along with Cdc42 in Diaph2-ve cells. Did the authors investigate whether RhoA activity was also impacted by HIV’s highjack of the actin remodeling machinery? This could be important because donor cell rounding appears to precede the final stages of viral transfer and RhoA-mediated activation of actomyosin contractility is required for cell rounding.

Minor points:

 - Statistics on Fig 4F: it is unclear to which condition the different samples were compared to.

- A plot showing the quantification of Arp2/3 distance to the filopodial tip (lines 196) would be important to see how these varied among the 50 cells(?)/filopodia(?) analyzed.

 - Fig. 6: The untreated (parental?) cell image should  be replaced by one also showing non-infected cells!

Author Response

Specific comments:

1) The significance of this study is limited by the fact that all conclusions on the impact of HIV in actin remodeling were withdrawn from experiments with clones isolated from one single immortalized monocytic cell line - U-937. Given that there is recent work that proposes that IQGAP1 signaling negatively regulates HIV trafficking, budding and virion egress, it would be important for the authors to show that their observations are not exclusive to the cell model utilized. It would be valuable if the authors could demonstrate the relevance of Cdc42 and Rac1 signaling for viral egress and cell-cell transfer in primary cells, namely by using chemical inhibitors such as EHT1864 and ML141, or even the Arp2/3 inhibitor CK-666.

The cell line used was specifically selected to mechanistically dissect a phenotype that we have previously observed and extensively characterized in primary immature dendritic cells in the context of HIV infection (i.e. Arp2/3-Diaph2 dependent filopodia associated with HIV buds). Importantly, these structures are also observed in vivo 14. So in this setting we consolidated and were stringent in cloning cells everytime we observed key phenotypes. As for the recent work that proposes negative regulation of Gag, yes it indeed maybe cell dependent. The key observation though is Gag is engaged in both settings by IQGAP1. As to the outcome, as we have mapped in our work, it will not only depend on the cell but also what the cell is doing with its F-Actin network at that specific point in time. So yes, Gag may indeed be negatively regulated when a specific F-Actin pathway is in action. It may indeed be beneficial for the virus, as Gag will not be engaged at a time that is not beneficial for viral spread. This is outlined in the discussion as follows:

“For instance near complete removal of IQGAP1 did not increase viral budding and egress, but rather led to inhibition thereof.  In light of our observations herein and those recently published 15, it can be concluded that given IQGAP1 is a scaffolding protein with many binding partners, the fate of IQGAP1 bound HIV Gag maybe many. Furthermore, these fates will depend on each cell type it is expressed in and what functions that cell type maybe engaged over the time the cells were sampled. Importantly, we do readily support a role for IQGAP1 in the viral life cycle and this readily supports recent observations by this team. “

With regard to the use of cytoskeletal inhibitors in primary dendritic cells. Yes, we have tested these compounds and unfortunately do not observe a phenotype a drug levels that do not lead to significant cellular toxicities. This is primarily why we have used our approach herein with this model cell line.  To provide this insight in the manuscript we have added the following statement in the results section:

“To confirm the role of Arp2/3 in this process we attempted to disable the complex with the small-molecule inhibitor CK-666.  However, use of this compound led to extensive toxicity at doses >200 uM and no effect on cell-cell HIV transfer was observed at lower doses (data not shown).  To explore which Arp2/3-dependent pathways were involved in this process, we instead independently disrupted the Rac1-WAVE2 pathway (lamellipodia) and the Cdc42-IQGAP1 pathway (filopodia).  “

2) Still regarding the Arp2/3 complex, since the authors argue the importance of its activation downstream of the HIV-stimulated Cdc42-IQGAP and Rac1-Wave2 pathways, how do they justify that there is no appreciable decrease in viral transfer nor in virion release in Diaph2-ve+Arp2/3-ve cells (re: Fig7)?

Although the reduction in cell-cell transfer by Diaph2-Arp2/3 double knockdown cells was modest, it was statistically significant in our experiments.  Since the Arp2/3 complex is enormously abundant and essential for a multitude of critical cellular processes, we preferred to limit the study of this double-knockdown cell line to morphological and topological features.  For functional phenotypes, such as the cellular ability to mediate cell-cell viral transfer, we instead focused on cell lines that were selectively deficient in specific Rho-GTPases (acting upstream of the Arp2/3 complex) or nucleation promoting factors (which assist the Arp2/3 complex), as is common practice in the literature.

3) Still on this note, the discussion should be streamlined and speculative sentences and conclusions avoided. For example, in lines 474-475 – the authors state that “(…) with IQGAP1 recruited by HIV Gag, Cdc42 would be maintained in an active GTP bound state [43]”, however, to my knowledge, there is no evidence, and certainly not in the cited reference, that binding to IQGAP can maintain Cdc42 in an active conformation. To sustain this claim, the authors would have to use CRIB-pulldown assays to demonstrate this locking of Cdc42 in the GTP-bound state after HIV gag-mediated recruitment of IQGAP upon Rac1 depletion.

The reference cited refers to work that demonstrates IQGAP1 oligomerisation when CDC42 is in its GTP bound state. Work by the Sacks lab 16 has observed IQGAP1 to maintain CDC42 in a primarily GTP bound state. In this same work, overexpression of IQGAP1deltaGRD led to lower levels of GTP bound CDC42 in cell lysates. We have added this reference to avoid any confusion.

4) Another example can be seen in lines 483-484: “(…) we hypothesise leading edge structures positive with HIV can indeed indirectly contribute to cell-cell spread and that only a proportion of HIV buds at the membrane engage in this process.” Then, shouldn't this fail in cells deprived of lamellipodia through Rac1/Wave depletion?

This confusion is likely derived from what we term leading edge structures vs. what historical literature refers to as the “leading edge”. We refer to the tips of filopodia and the crests of lamellipodia as “leading edge”, however we have re-worded the discussion and limited use of this terminology to avoid potential confusion.

5) In the HIV egress proteomic analysis the abundance of RhoA also increased along with Cdc42 in Diaph2-ve cells. Did the authors investigate whether RhoA activity was also impacted by HIV’s highjack of the actin remodeling machinery? This could be important because donor cell rounding appears to precede the final stages of viral transfer and RhoA-mediated activation of actomyosin contractility is required for cell rounding.

Not appearing in this study is an extensive look at over 50 cytoskeletal regulators and their role in cell-cell spread. We have explored a potential role of RhoA and not found any evidence that disruption of this gene in the donor cell reduces cell-cell transmission.  The role of RhoA in the target cell is being addressed as a separate project that is beyond the scope of this manuscript.  To add some insight re. RhoA in this work, we have highlighted in the mass-spectrometry result section that RhoA is detected alongside Cdc42 and Rac1.  We have also added the following sentence to the results discussion:  

 Of note, we also initially explored a potential role of RhoA, as it was identified alongside Cdc42 and Rac1 in the virion proteome (Fig. 5 A&B).  However, given a complete lack of morphological or cell-cell transfer phenotype in RhoA-depleted donor cells (data not shown), and little literature evidence of RhoA-mediated Arp2/3 regulation, this was not pursued further. 

Minor points:

 - Statistics on Fig 4F: it is unclear to which condition the different samples were compared to.

The number of filopodia per cell for each condition have been compared to cells infected with WT HIViGag.

- A plot showing the quantification of Arp2/3 distance to the filopodial tip (lines 196) would be important to see how these varied among the 50 cells(?)/filopodia(?) analyzed.

The data has now been included as supplementary figure 4

 - Fig. 6: The untreated (parental?) cell image should be replaced by one also showing non-infected cells!

Fixed cell imaging was done using 100 x 1.4 NA objective and the primary objective herein was to capture an infected cell with its extensive filopodial network in entirety. For sake of clarity we have now included an image of an uninfected cell as an inset in Fig 6, panel A.

References cited within this cover letter

  1. Aggarwal A, Iemma TL, Shih I, et al. Mobilization of HIV spread by diaphanous 2 dependent filopodia in infected dendritic cells. PLoS pathogens. 2012;8(6):e1002762.
  2. Nikolic DS, Lehmann M, Felts R, et al. HIV-1 activates Cdc42 and induces membrane extensions in immature dendritic cells to facilitate cell-to-cell virus propagation. Blood. 2011;118(18):4841-4852.
  3. Sundstrom C, Nilsson K. Establishment and characterization of a human histiocytic lymphoma cell line (U-937). Int J Cancer. 1976;17(5):565-577.
  4. Liu L, Zubik L, Collins FW, Marko M, Meydani M. The antiatherogenic potential of oat phenolic compounds. Atherosclerosis. 2004;175(1):39-49.
  5. Liu H, Fang S, Wang W, et al. Macrophage-derived MCPIP1 mediates silica-induced pulmonary fibrosis via autophagy. Part Fibre Toxicol. 2016;13(1):55.
  6. Ponten F, Jirstrom K, Uhlen M. The Human Protein Atlas--a tool for pathology. J Pathol. 2008;216(4):387-393.
  7. Wu C, Jin X, Tsueng G, Afrasiabi C, Su AI. BioGPS: building your own mash-up of gene annotations and expression profiles. Nucleic Acids Res. 2016;44(D1):D313-316.
  8. Eugenin EA, Gaskill PJ, Berman JW. Tunneling nanotubes (TNT) are induced by HIV-infection of macrophages: a potential mechanism for intercellular HIV trafficking. Cellular immunology. 2009;254(2):142-148.
  9. Bourinbaiar AS, Phillips DM. Transmission of human immunodeficiency virus from monocytes to epithelia. Journal of acquired immune deficiency syndromes. 1991;4(1):56-63.
  10. Pearce-Pratt R, Malamud D, Phillips DM. Role of the cytoskeleton in cell-to-cell transmission of human immunodeficiency virus. Journal of virology. 1994;68(5):2898-2905.
  11. Felts RL, Narayan K, Estes JD, et al. 3D visualization of HIV transfer at the virological synapse between dendritic cells and T cells. Proceedings of the National Academy of Sciences of the United States of America. 2010;107(30):13336-13341.
  12. Mattila PK, Lappalainen P. Filopodia: molecular architecture and cellular functions. Nature reviews Molecular cell biology. 2008;9(6):446-454.
  13. Shrivastava A, Prasad A, Kuzontkoski PM, Yu J, Groopman JE. Slit2N Inhibits Transmission of HIV-1 from Dendritic Cells to T-cells by Modulating Novel Cytoskeletal Elements. Sci Rep. 2015;5:16833.
  14. Ladinsky MS, Kieffer C, Olson G, et al. Electron tomography of HIV-1 infection in gut-associated lymphoid tissue. PLoS pathogens. 2014;10(1):e1003899.
  15. Sabo Y, de Los Santos K, Goff SP. IQGAP1 Negatively Regulates HIV-1 Gag Trafficking and Virion Production. Cell Rep. 2020;30(12):4065-4081 e4064.
  16. Swart-Mataraza JM, Li Z, Sacks DB. IQGAP1 is a component of Cdc42 signaling to the cytoskeleton. The Journal of biological chemistry. 2002;277(27):24753-24763.

Round 2

Reviewer 2 Report

The authors have responded well to the concerns raised and improved the clarity of the manuscript. Although the additional experiments suggested would further strengthen their conclusion, it is understandable they would be technically challenging and time consuming.

Reviewer 3 Report

The authors improved the clarity of the manuscript by addressing most of the issues raised.